Published in Transations on Machine Learning Research (07/2022)

# Learning to Switch Among Agents in a Team via $2$-Layer Markov Decision Processes

**Vahid Balazadeh**                                           *vahid@cs.toronto.edu*
*University of Toronto*

**Abir De**                                                   *abir@cse.iitb.ac.in*
*Indian Institute of Technology Bombay*

**Adish Singla**                                             *adishs@mpi-sws.dot.org*
*Max Planck Institute for Software Systems*

**Manuel Gomez Rodriguez**                                   *manuelgr@mpi-sws.org*
*Max Planck Institute for Software Systems*

**Reviewed on OpenReview:** *https://openreview.net/forum?id=NT9zgedd3I*

## Abstract

Reinforcement learning agents have been mostly developed and evaluated under the assumption that they will operate in a fully autonomous manner—they will take *all* actions. In this work, our goal is to develop algorithms that, by learning to switch control between agents, allow existing reinforcement learning agents to operate under different automation levels. To this end, we first formally define the problem of learning to switch control among agents in a team via a 2-layer Markov decision process. Then, we develop an online learning algorithm that uses upper confidence bounds on the agents' policies and the environment's transition probabilities to find a sequence of switching policies. The total regret of our algorithm with respect to the optimal switching policy is sublinear in the number of learning steps and, whenever multiple teams of agents operate in a similar environment, our algorithm greatly benefits from maintaining shared confidence bounds for the environments' transition probabilities and it enjoys a better regret bound than problem-agnostic algorithms. Simulation experiments illustrate our theoretical findings and demonstrate that, by exploiting the specific structure of the problem, our proposed algorithm is superior to problem-agnostic algorithms.

## 1 Introduction

In recent years, reinforcement learning (RL) agents have achieved, or even surpassed, human performance in a variety of computer games by taking decisions autonomously, without human intervention (Mnih et al., 2015; Silver et al., 2016; 2017; Vinyals et al., 2019). Motivated by these successful stories, there has been a tremendous excitement on the possibility of using RL agents to operate fully autonomous cyberphysical systems, especially in the context of autonomous driving. Unfortunately, a number of technical, societal, and legal challenges have precluded this possibility to become so far a reality.

In this work, we argue that existing RL agents may still enhance the operation of cyberphysical systems if deployed under lower automation levels. For example, if we let RL agents take some of the actions and leave the remaining ones to human agents, the resulting performance may be better than the performance either of them would achieve on their own (Raghu et al., 2019a; De et al., 2020; Wilder et al., 2020). Once we depart from full automation, we need to address the following question: when should we switch control between machine and human agents? In this work, we look into this problem from a theoretical perspective and develop an online algorithm that learns to optimally switch control between multiple agents in a team automatically. However, to fulfill this goal, we need to address several challenges:

— *Level of automation.* In each application, what is considered an appropriate and tolerable load for each agent may differ (European Parliament, 2006). Therefore, we would like that our algorithms provide mechanisms to adjust the amount of control for each agent (*i.e.*, level of automation) during a given time period.

— *Number of switches.* Consider two different switching patterns resulting in the same amount of agent control and equivalent performance. Then, we would like our algorithms to favor the pattern with the least number of switches. For example, in a team consisting of human and machine agents, every time a machine defers (takes) control to (from) a human, there is an additional cognitive load for the human (Brookhuis et al., 2001).

— *Unknown agent policies.* The spectrum of human abilities spans a broad range (Macadam, 2003). As a result, there is a wide variety of potential human policies. Here, we would like that our algorithms learn personalized switching policies that, over time, adapt to the particular humans (and machines) they are dealing with.

— *Disentangling agents' policies and environment dynamics.* We would like that our algorithms learn to disentangle the influence of the agents' policies and the environment dynamics on the switching policies. By doing so, they could be used to efficiently find multiple personalized switching policies for different teams of agents operating in similar environments (*e.g.*, multiple semi-autonomous vehicles with different human drivers).

To tackle the above challenges, we first formally define the problem of learning to switch control among agents in a team using a 2-layer Markov decision process (Figure 1). Here, the team can be composed of any number of machines or human agents, and the agents' policies, as well as the transition probabilities of the environment, may be unknown. In our formulation, we assume that all agents follow Markovian policies[1], similarly as other theoretical models of human decision making (Townsend et al., 2000; Daw & Dayan, 2014; McGhan et al., 2015). Under this definition, the problem reduces to finding the switching policy that provides an optimal trade off between the environmental cost, the amount of agent control, and the number of switches. Then, we develop an online learning algorithm, which we refer to as UCRL2-MC[2], that uses upper confidence bounds on the agents' policies and the transition probabilities of the environment to find a sequence of switching policies whose total regret with respect to the optimal switching policy is sublinear in the number of learning steps. In addition, we also demonstrate that the same algorithm can be used to find multiple sequences of switching policies across several independent teams of agents operating in similar environments, where it greatly benefits from maintaining shared confidence bounds for the transition probabilities of the environments and enjoys a better regret bound than UCRL2, a very well known reinforcement learning algorithm that we view as the most natural competitor. Finally, we perform a variety of simulation experiments in the standard RiverSwim environment as well as an obstacle avoidance task, where we consider multiple teams of agents (drivers) composed by one human and one machine agent. Our results illustrate our theoretical findings and demonstrate that, by exploiting the specific structure of the problem, our proposed algorithm is superior to problem-agnostic alternatives.

Before we proceed further, we would like to point out that, at a broader level, our methodology and theoretical results are applicable to the problem of switching control between agents following Markovian policies. As long as the agent policies are Markovian, our results do not distinguish between machine and human agents. In this context, we view teams of human and machine agents as one potential application of our work, which we use as a motivating example throughout the paper. However, we would also like to acknowledge that a practical deployment of our methodology in a real application with human and machine agents would require considering a wide range of additional practical aspects (*e.g.*, transparency, explainability, and visualization). Moreover, one may also need to explicitly model the difference in reaction times between human and machine agents. Finally, there may be scenarios in which it might be beneficial to allow a human operator to switch control. Such considerations are out of the scope of our work.

---

[1]In certain cases, it is possible to convert a non-Markovian human policy into a Markovian one by changing the state representation (Daw & Dayan, 2014). Addressing the problem of learning to switch control among agents in a team in a semi-Markovian setting is left as a very interesting venue for future work.

[2]UCRL2 with Multiple Confidence sets.

## 2 Related work

One can think of applying existing RL algorithms (Jaksch et al., 2010; Osband et al., 2013; Osband & Van Roy, 2014; Gopalan & Mannor, 2015), such as UCRL2 or Rmax, to find switching policies. However, these problem-agnostic algorithms are unable to exploit the specific structure of our problem. More specifically, our algorithm computes the confidence intervals separately over the agents' policies and the transition probabilities of the environment, instead of computing a single confidence interval, as problem-agnostic algorithms do. As a consequence, our algorithm learns to switch more efficiently across multiple teams of agents, as shown in Section 6.

There is a rapidly increasing line of work on learning to defer decisions in the machine learning literature (Bartlett & Wegkamp, 2008; Cortes et al., 2016; Geifman et al., 2018; Ramaswamy et al., 2018; Geifman & El-Yaniv, 2019; Liu et al., 2019; Raghu et al., 2019a;b; Thulasidasan et al., 2019; De et al., 2020; 2021; Mozannar & Sontag, 2020; Wilder et al., 2020; Shekhar et al., 2021). However, previous work has typically focused on supervised learning. More specifically, it has developed classifiers that learn to defer by considering the defer action as an additional label value, by training an independent classifier to decide about deferred decisions, or by reducing the problem to a combinatorial optimization problem. Moreover, except for a few recent notable exceptions (Raghu et al., 2019a; De et al., 2020; 2021; Mozannar & Sontag, 2020; Wilder et al., 2020), they do not consider there is a human decision maker who takes a decision whenever the classifiers defer it. In contrast, we focus on reinforcement learning, and develop algorithms that learn to switch control between multiple agents, including human agents. Recently, Jacq et al. (2022) introduced a new framework called lazy-MDPs to decide when to act optimally for reinforcement learning agents. They propose to augment existing MDPs with a new default action and encourage agents to defer decision-making to default policy in non-critical states. Though their lazy-MDP is similar to our augmented 2-layer MDP framework, our approach is designed to switch optimally between possibly multiple agents, each having its own policy.

Our work is also connected to research on understanding switching behavior and switching costs in the context of human-computer interaction (Czerwinski et al., 2000; Horvitz & Apacible, 2003; Iqbal & Bailey, 2007; Kotowick & Shah, 2018; Janssen et al., 2019), which has been sometimes referred to as "adjustable autonomy" (Mostafa et al., 2019). At a technical level, our work advances state of the art in adjustable autonomy by introducing an algorithm with provable guarantees to efficiently find the optimal switching policy in a setting in which the dynamics of the environment and the agents' policies are unknown (*i.e.*, there is uncertainty about them). Moreover, our work also relates to a recent line of research that combines deep reinforcement learning with opponent modeling to robustly switch between multiple machine policies (Everett & Roberts, 2018; Zheng et al., 2018). However, this line of research does not consider the presence of human agents, and there are no theoretical guarantees on the performance of the proposed algorithms.

Furthermore, our work contributes to an extensive body of work on human-machine collaboration (Stone et al., 2010; Taylor et al., 2011; Walsh et al., 2011; Barrett & Stone, 2012; Macindoe et al., 2012; Torrey & Taylor, 2013; Nikolaidis et al., 2015; Hadfield-Menell et al., 2016; Nikolaidis et al., 2017; Grover et al., 2018; Haug et al., 2018; Reddy et al., 2018; Wilson & Daugherty, 2018; Brown & Niekum, 2019; Kamalaruban et al., 2019; Radanovic et al., 2019; Tschiatschek et al., 2019; Ghosh et al., 2020; Strouse et al., 2021). However, rather than developing algorithms that learn to switch control between humans and machines, previous work has predominantly considered settings in which the machine and the human interact with each other.

Finally, one can think of using option framework and the notion of macro-actions and micro-actions to formulate the problem of learning to switch (Sutton et al., 1999). However, the option framework is designed to address different levels of temporal abstraction in RL by defining macro-actions that correspond to sub-tasks (skills). In our problem, each agent is not necessarily optimized to act for a specific task or sub-goal but for the whole environment/goal. Also, in our problem, we do not necessarily have control over all agents to learn the optimal policy for each agent, while in the option framework, a primary direction is to learn optimal options for each sub-task. In other words, even though we can mathematically refer to each agent policy as an option, they are not conceptually the same.

## 3 Switching Control Among Agents as a 2-Layer MDP

Given a team of agents $\mathcal{D}$, at each time step $t \in \{1, \ldots, L\}$, our (cyberphysical) system is characterized by a state $s_t \in \mathcal{S}$, where $\mathcal{S}$ is a finite state space, and a control switch $d_t \in \mathcal{D}$, which determines who takes an action $a_t \in \mathcal{A}$, where $\mathcal{A}$ is a finite action space. In the above, the switch value is given by a (deterministic and time-varying) switching policy $d_t = \pi_t(s_t, d_{t-1})$[3]. More specifically, if $d_t = d$, the action $a_t$ is sampled from the agent $d$'s policy $p_d(a_t \,|\, s_t)$. Moreover, given a state $s_t$ and an action $a_t$, the state $s_{t+1}$ is sampled from a transition probability $p(s_{t+1} \,|\, s_t, a_t)$. Here, we assume that the agents' policies and the transition probabilities may be unknown. Finally, given an initial state and switch value $(s_1, d_0)$ and a trajectory $\tau = \{(s_t, d_t, a_t)\}_{t=1}^{L}$ of states, switch values and actions, we define the total cost $c(\tau \,|\, s_1, d_0)$ as:

$$c(\tau \,|\, s_1, d_0) = \sum_{t=1}^{L} [c_e(s_t, a_t) + c_c(d_t) + c_x(d_t, d_{t-1})], \tag{1}$$

where $c_e(s_t, a_t)$ is the environment cost of taking action $a_t$ at state $s_t$, $c_c(d_t)$ is the cost of giving control to agent $d_t$, $c_x(d_t, d_{t-1})$ is the cost of switching from $d_{t-1}$ to $d_t$, and $L$ is the time horizon[4]. Then, our goal is to find the optimal switching policy $\pi^* = (\pi_1^*, \ldots, \pi_L^*)$ that minimizes the expected cost, *i.e.*,

$$\pi^* = \operatorname*{argmin}_{\pi} \mathbb{E}\left[c(\tau \,|\, s_1, d_0)\right], \tag{2}$$

where the expectation is taken over all the trajectories induced by the switching policy given the agents' policies.

To solve the above problem, one could just resort to problem-agnostic RL algorithms, such as UCRL2 or Rmax, over a standard Markov decision process (MDP), defined as

$$\mathcal{M} = (\mathcal{S} \times \mathcal{D}, \mathcal{D}, \bar{P}, \bar{C}, L),$$

where $\mathcal{S} \times \mathcal{D}$ is an augmented state space, the set of actions $\mathcal{D}$ is just the switch values, the transition dynamics $\bar{P}$ at time $t$ are given by

$$p(s_{t+1}, d_t \,|\, s_t, d_{t-1}) = \mathbb{I}[\pi_t(s_t, d_{t-1}) = d_t] \times \sum_{a \in \mathcal{A}} p(s_{t+1} \,|\, s_t, a) p_{d_t}(a \,|\, s_t), \tag{3}$$

the immediate cost $\bar{C}$ at time $t$ is given by

$$\bar{c}(s_t, d_{t-1}) = \mathbb{E}_{a_t \sim p_{\pi_t(s_t, d_{t-1})}(\cdot \,|\, s_t)} [c_e(s_t, a_t)] + c_c(\pi_t(s_t, d_{t-1})) + c_x(\pi_t(s_t, d_{t-1}), d_{t-1}). \tag{4}$$

Here, note that, by using conditional expectations, we can compute the average cost of a trajectory, given by Eq. 1, from the above immediate costs. However, these algorithms would not exploit the structure of the problem. More specifically, they would not use the observed agents' actions to improve the estimation of the transition dynamics over time.

To avoid the above shortcoming, we will resort instead to a 2-layer MDP where taking an action $d_t$ in state $(s_t, d_{t-1})$ leads first to an intermediate state $(s_t, a_t) \in \mathcal{S} \times \mathcal{A}$ with probability $p_{d_t}(a_t \,|\, s_t)$ and immediate cost $c_{d_t}(s_t, d_{t-1}) = c_c(d_t) + c_x(d_t, d_{t-1})$ and then to a final state $(s_{t+1}, d_t) \in \mathcal{S} \times \mathcal{D}$ with probability $\mathbb{I}[\pi_t(s_t, d_{t-1}) = d_t] \cdot p(s_{t+1} \,|\, s_t, a_t)$ and immediate cost $c_e(s_t, a_t)$. More formally, the 2-layer MDP is defined by the following 8-tuple:

$$\mathcal{M} = (\mathcal{S} \times \mathcal{D}, \mathcal{S} \times \mathcal{A}, \mathcal{D}, P_{\mathcal{D}}, P, C_{\mathcal{D}}, C_e, L) \tag{5}$$

where $\mathcal{S} \times \mathcal{D}$ is the final state space, $\mathcal{S} \times \mathcal{A}$ is the intermediate state space, the set of actions $\mathcal{D}$ is the switch values, the transition dynamics $P_{\mathcal{D}}$ and $P$ at time $t$ are given by $p_{d_t}(a_t \,|\, s_t)$ and $\mathbb{I}[\pi_t(s_t, d_{t-1}) = d_t] \cdot p(s_{t+1} \,|\, s_t, a_t)$, and the immediate costs $C_{\mathcal{D}}$ and $C_e$ at time $t$ are given by $c_{d_t}(s_t, d_{t-1})$ and $c_e(s_t, a_t)$, respectively.

The above 2-layer MDP will allow us to estimate separately the agents' policies $p_d(\cdot \,|\, s)$ and the transition probability $p(\cdot \,|\, s, a)$ of the environment using both the intermediate and final states and design an algorithm that improves the regret that problem-agnostic RL algorithms achieve in our problem.

---

[3]Note that, by making the switching policy dependent on the previous switch value $d_{t-1}$, we can account for the switching cost.

[4]The specific choice of environment cost $c_e(\cdot, \cdot)$, control cost $c_c(\cdot)$ and switching cost $c_x(\cdot, \cdot)$ is application dependent.

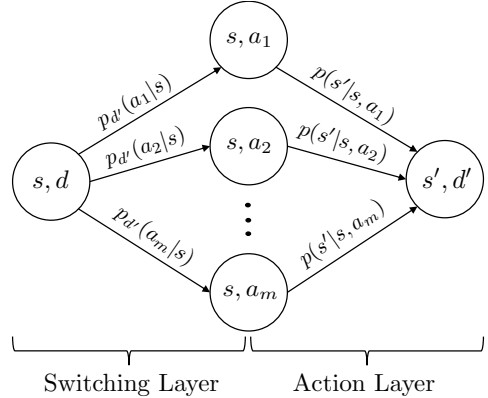

Figure 1: Transitions of a 2-layer Markov Decision Process (MDP) from state $(s, d)$ to state $(s', d')$ after seleting agent $d'$. $d'$ and $d$ denote the current and previous agents in control. In the first layer (switching layer), the switching policy chooses agent $d'$, which takes action w.r.t. its action policy $p_{d'}$. Then, in the action layer, the environment transitions to the next state $s'$ based on the taken action w.r.t. the transition probability $p$.

## 4  Learning to Switch in a Team of Agents

Since we may not know the agents' policies nor the transition probabilities, we need to trade off exploitation, *i.e.*, minimizing the expected cost, and exploration, *i.e.*, learning about the agents' policies and the transition probabilities. To this end, we look at the problem from the perspective of episodic learning and proceed as follows.

We consider $K$ independent subsequent episodes of length $L$ and denote the aggregate length of all episodes as $T = KL$. Each of these episodes corresponds to a realization of the same finite horizon 2-layer Markov decision process, introduced in Section 3, with state spaces $\mathcal{S} \times \mathcal{A}$ and $\mathcal{S} \times \mathcal{D}$, set of actions $\mathcal{D}$, true agent policies $P_{\mathcal{D}}^*$, true environment transition probability $P^*$, and immediate costs $C_{\mathcal{D}}$ and $C_e$. However, since we do not know the true agent policies and environment transition probabilities, just before each episode $k$ starts, our goal is to find a switching policy $\pi^k$ with desirable properties in terms of total regret $R(T)$, which is given by:

$$R(T) = \sum_{k=1}^{K} \left[ \mathbb{E}_{\tau \sim \pi^k, P_{\mathcal{D}}^*, P^*} \left[ c(\tau \mid s_1, d_0) \right] - \mathbb{E}_{\tau \sim \pi^*, P_{\mathcal{D}}^*, P^*} \left[ c(\tau \mid s_1, d_0) \right] \right], \tag{6}$$

where $\pi^*$ is the optimal switching policy under the true agent policies and environment transition probabilities.

To achieve our goal, we apply the principle of *optimism in the face of uncertainty*, *i.e.*,

$$\pi^k = \operatorname*{argmin}_{\pi} \min_{P_{\mathcal{D}} \in \mathcal{P}_{\mathcal{D}}^k} \min_{P \in \mathcal{P}^k} \mathbb{E}_{\tau \sim \pi, P_{\mathcal{D}}, P} \left[ c(\tau \mid s_1, d_0) \right] \tag{7}$$

where $\mathcal{P}_{\mathcal{D}}^k$ is a $(|\mathcal{S}| \times |\mathcal{D}| \times L)$-rectangular confidence set, *i.e.*, $\mathcal{P}_{\mathcal{D}}^k = \bigtimes_{s,d,t} \mathcal{P}_{\cdot \mid d,s,t}^k$, and $\mathcal{P}^k$ is a $(|\mathcal{S}| \times |\mathcal{A}| \times L)$-rectangular confidence set, *i.e.*, $\mathcal{P}^k = \bigtimes_{s,a,t} \mathcal{P}_{\cdot \mid s,a,t}^k$. Here, note that the confidence sets are constructed using data gathered during the first $k - 1$ episodes and allows for time-varying agent policies $p_d(\cdot \mid s, t)$ and transition probabilities $p(\cdot \mid s, a, t)$.

However, to solve Eq. 7, we first need to explicitly define the confidence sets. To this end, we first define the empirical distributions $\hat{p}_d^k(\cdot \mid s)$ and $\hat{p}^k(\cdot \mid s, a)$ just before episode $k$ starts as:

$$\hat{p}_d^k(a \mid s) = \begin{cases} \frac{N_k(s,d,a)}{N_k(s,d)} & \text{if } N_k(s,d) \neq 0 \\ \frac{1}{|\mathcal{A}|} & \text{otherwise,} \end{cases} \tag{8}$$

$$\hat{p}^k(s' \mid s, a) = \begin{cases} \frac{N_k'(s,a,s')}{N_k'(s,a)} & \text{if } N_k'(s,a) \neq 0 \\ \frac{1}{|\mathcal{S}|} & \text{otherwise,} \end{cases} \tag{9}$$

where

$$N_k(s,d) = \sum_{l=1}^{k-1} \sum_{t \in [L]} \mathbb{I}(s_t = s, d_t = d \text{ in episode } l), \ N_k(s,d,a) = \sum_{l=1}^{k-1} \sum_{t \in [L]} \mathbb{I}(s_t = s, a_t = a, d_t = d \text{ in episode } l),$$

$$N_k'(s,a) = \sum_{l=1}^{k-1} \sum_{t \in [L]} \mathbb{I}(s_t = s, a_t = a \text{ in episode } l), \ N_k'(s,a,s') = \sum_{l=1}^{k-1} \sum_{t \in [L]} \mathbb{I}(s_t = s, a_t = a, s_{t+1} = s' \text{ in episode } l).$$

Then, similarly as in Jaksch et al. (2010), we opt for $L^1$ confidence sets[5], *i.e.*,

$$\mathcal{P}_{\cdot \mid d,s,t}^k(\delta) = \left\{ p_d : ||p_d(\cdot \mid s, t) - \hat{p}_d^k(\cdot \mid s)||_1 \leq \beta_{\mathcal{D}}^k(s,d,\delta) \right\},$$

$$\mathcal{P}_{\cdot \mid s,a,t}^k(\delta) = \left\{ p : ||p(\cdot \mid s, a, t) - \hat{p}^k(\cdot \mid s, a)||_1 \leq \beta^k(s,a,\delta) \right\},$$

for all $d \in \mathcal{D}$, $s \in \mathcal{S}$, $a \in \mathcal{A}$ and $t \in [L]$, where $\delta$ is a given parameter,

$$\beta_{\mathcal{D}}^k(s,d,\delta) = \sqrt{\frac{2\log\left(\frac{(k-1)^7 L^7 |\mathcal{S}||\mathcal{D}|2^{|\mathcal{A}|+1}}{\delta}\right)}{\max\{1, N_k(s,d)\}}} \quad \text{and} \quad \beta^k(s,a,\delta) = \sqrt{\frac{2\log\left(\frac{(k-1)^7 L^7 |\mathcal{S}||\mathcal{A}|2^{|\mathcal{S}|+1}}{\delta}\right)}{\max\{1, N_k(s,a)\}}}.$$

Next, given the switching policy $\pi$ and the transition dynamics $P_{\mathcal{D}}$ and $P$, we define the value function as

$$V_{t|P_{\mathcal{D}},P}^\pi(s,d) = \mathbb{E}\left[\sum_{\tau=t}^L c_e(s_\tau, a_\tau) + c_c(d_\tau) + c_x(d_\tau, d_{\tau-1}) \mid s_t = s, d_{t-1} = d\right], \tag{10}$$

where the expectation is taken over all the trajectories induced by the switching policy given the agents' policies. Then, for each episode $k$, we define the optimal value function $v_t^k(s,d)$ as

$$v_t^k(s,d) = \min_\pi \min_{P_{\mathcal{D}} \in \mathcal{P}_{\mathcal{D}}^k(\delta)} \min_{P \in \mathcal{P}^k(\delta)} V_{t|P_{\mathcal{D}},P}^\pi(s,d). \tag{11}$$

Then, we are ready to use the following key theorem, which gives a solution to Eq. 7 (proven in Appendix A):

**Theorem 1.** *For any episode $k$, the optimal value function $v_t^k(s,d)$ satisfies the following recursive equation:*

$$v_t^k(s,d) = \min_{d_t \in \mathcal{D}} \left[ c_{d_t}(s,d) + \min_{p_{d_t} \in \mathcal{P}_{\cdot \mid d_t,s,t}^k} \sum_{a \in \mathcal{A}} p_{d_t}(a \mid s, t) \times \left( c_e(s,a) + \min_{p \in \mathcal{P}_{\cdot \mid s,a,t}^k} \mathbb{E}_{s' \sim p(\cdot \mid s,a,t)}[v_{t+1}^k(s', d_t)] \right) \right], \tag{12}$$

*with $v_{L+1}^k(s,d) = 0$ for all $s \in \mathcal{S}$ and $d \in \mathcal{D}$. Moreover, if $d_t^*$ is the solution to the minimization problem of the RHS of the above recursive equation, then $\pi_t^k(s,d) = d_t^*$.*

The above result readily implies that, just before each episode $k$ starts, we can find the optimal switching policy $\pi^k = (\pi_1^k, \ldots, \pi_L^k)$ using dynamic programming, starting with $v_{L+1}(s,d) = 0$ for all $s \in \mathcal{S}$ and $d \in \mathcal{D}$. Moreover, similarly as in Strehl & Littman (2008), we can solve the inner minimization problems in Eq. 12 analytically using Lemma 7 in Appendix B. To this end, we first find the optimal $p(\cdot \mid s, a, t)$ for all and $a \in \mathcal{A}$

---

[5]This choice will result into a sequence of switching policies with desirable properties in terms of total regret.

---

**ALGORITHM 1:** UCRL2-MC

---
1: Cost functions $C_{\mathcal{D}}$ and $C_e$, $\delta$
2: $\{N_k, N_k'\} \leftarrow$ INITIALIZECOUNTS()
3: **for** $k = 1, \ldots, K$ **do**
4:     $\{\hat{p}_d^k\}, \hat{p}^k \leftarrow$ UPDATEDISTRIBUTION($\{N_k, N_k'\}$)
5:     $\mathcal{P}_{\mathcal{D}}^k, \mathcal{P}^k \leftarrow$ UPDATECONFIDENCESETS($\{\hat{p}_d^k\}, \hat{p}^k, \delta$)
6:     $\pi^k \leftarrow$ GETOPTIMAL($\mathcal{P}_{\mathcal{D}}^k, \mathcal{P}^k, C_{\mathcal{D}}, C_e$),
7:     $(s_1, d_0) \leftarrow$ INITIALIZECONDITIONS()
8:     **for** $t = 1, \ldots, L$ **do**
9:         $d_t \leftarrow \pi_t^k(s_t, d_{t-1})$
10:         $a_t \sim p_{d_t}(\cdot \mid s_t)$
11:         $s_{t+1} \sim P(\cdot \mid s_t, a_t)$
12:         $\mathcal{N} \leftarrow$ UPDATECOUNTS($(s_t, d_t, a_t, s_{t+1}), \{N_k, N_k'\}$)
13:     **end for**
14: **end for**
15: **Return** $\pi^K$

---

and then we find the optimal $p_{d_t}(\cdot \mid s, t)$ for all $d_t \in \mathcal{D}$. Algorithm 1 summarizes the whole procedure, which we refer to as UCRL2-MC.

Within the algorithm, the function GETOPTIMAL($\cdot$) finds the optimal policy $\pi^k$ using dynamic programming, as described above, and UPDATEDISTRIBUTION($\cdot$) computes Eqs. 8 and 9. Moreover, it is important to notice that, in lines 8–10, the switching policy $\pi^k$ is actually deployed, the true agents take actions on the true environment and, as a result, action and state transition data from the true agents and the true environment is gathered.

Next, the following theorem shows that the sequence of policies $\{\pi^k\}_{k=1}^K$ found by Algorithm 1 achieve a total regret that is sublinear with respect to the number of steps, as defined in Eq. 6 (proven in Appendix A):

**Theorem 2.** *Assume we use Algorithm 1 to find the switching policies $\pi^k$. Then, with probability at least $1 - \delta$, it holds that*

$$R(T) \leq \rho_1 L \sqrt{|\mathcal{A}||\mathcal{S}||\mathcal{D}|T \log\left(\frac{|\mathcal{S}||\mathcal{D}|T}{\delta}\right)} + \rho_2 L |\mathcal{S}| \sqrt{|\mathcal{A}|T \log\left(\frac{|\mathcal{S}||\mathcal{A}|T}{\delta}\right)} \tag{13}$$

*where $\rho_1, \rho_2 > 0$ are constants.*

The above regret bound suggests that our algorithm may achieve higher regret than standard UCRL2 (Jaksch et al., 2010), one of the most popular problem-agnostic RL algorithms. More specifically, one can readily show that, if we use UCRL2 to find the switching policies $\pi^k$ (refer to Appendix C), then, with probability at least $1 - \delta$, it holds that

$$R(T) \leq \rho L |\mathcal{S}| \sqrt{|\mathcal{D}|T \log\left(\frac{|\mathcal{S}||\mathcal{D}|T}{\delta}\right)} \tag{14}$$

where $\rho$ is a constant. Then, if we omit constant and logarithmic factors and assume the size of the team of agents is smaller than the size of state space, *i.e.*, $|\mathcal{D}| < |\mathcal{S}|$, we have that, for UCRL2, the regret bound is $\tilde{\mathcal{O}}(L|\mathcal{S}|\sqrt{|\mathcal{D}|T})$ while, for UCRL2-MC, it is $\tilde{\mathcal{O}}(L|\mathcal{S}|\sqrt{|\mathcal{A}|T})$.

That being said, in practice, we have found that our algorithm achieves comparable regret with respect to UCRL2, as shown in Figure 4. In addition, after applying our algorithm on a specific team of agents and environment, we can reuse the confidence intervals over the transition probability $p(\cdot \mid s, a)$ we have learned to find the optimal switching policy for a different team of agents operating in a similar environment. In contrast, after applying UCRL2, we would only have a confidence interval over the conditional probability defined by Eq. 3, which would be of little use to find the optimal switching policy for a different team of agents.

In the following section, we will build up on this insight by considering several independent teams of agents operating in similar environments. We will demonstrate that, whenever we aim to find multiple sequences of

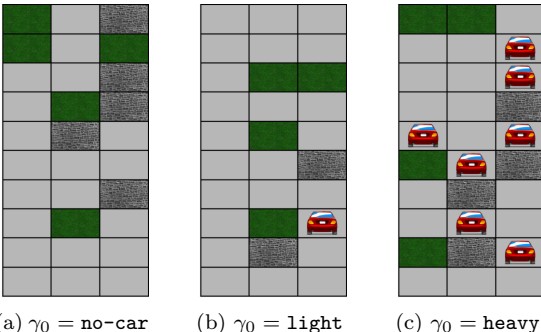

(a) $\gamma_0 = $ `no-car`    (b) $\gamma_0 = $ `light`    (c) $\gamma_0 = $ `heavy`

Figure 2: Three examples of environment realizations with different initial traffic level $\gamma_0$.

switching policies for these independent teams, a straightforward variation of UCRL2-MC greatly benefits from maintaining shared confidence bounds for the transition probabilities of the environments and enjoys a better regret bound than UCRL2.

**Remarks.** For ease of exposition, we have assumed that both the machine and human agents follow arbitrary Markov policies that do not change due to switching. However, our theoretical results still hold if we lift this assumption—we just need to define the agents' policies as $p_d(a_t|s_t, d_t, d_{t-1})$ and construct separate confidence sets based on the switch values.

## 5  Learning to Switch Across Multiple Teams of Agents

In this section, rather than finding a sequence of switching policies for a single team of agents, we aim to find multiple sequences of switching policies across several independent teams operating in similar environments. We will analyze our algorithm in scenarios where it can maintain shared confidence bounds for the transition probabilities of the environments across these independent teams. For instance, when the learning algorithm is deployed in centralized settings, it is possible to collect data across independent teams to maintain shared confidence intervals on the common parameters (i.e., the environment's transition probabilities in our problem setting). This setting fits a variety of real applications, more prominently, think of a car manufacturer continuously collecting driving data from million of human drivers wishing to learn different switching policies for each driver to implement a personalized semi-autonomous driving system. Similarly as in the previous section, we look at the problem from the perspective of episodic learning and proceed as follows.

Given $N$ independent teams of agents $\{\mathcal{D}_i\}_{i=1}^N$, we consider $K$ independent subsequent episodes of length $L$ per team and denote the aggregate length of all of these episodes as $T = KL$. For each team of agents $\mathcal{D}_i$, every episode corresponds to a realization of a finite horizon 2-layer Markov decision process with state spaces $\mathcal{S} \times \mathcal{A}$ and $\mathcal{S} \times \mathcal{D}_i$, set of actions $\mathcal{D}_i$, true agent policies $P_{\mathcal{D}_i}^*$, true environment transition probability $P^*$, and immediate costs $C_{\mathcal{D}_i}$ and $C_e$. Here, note that all the teams operate in a similar environment, *i.e.*, $P^*$ is shared across teams, and, without loss of generality, they share the same costs. Then, our goal is to find the switching policies $\pi_i^k$ with desirable properties in terms of total regret $R(T, N)$, which is given by:

$$R(T, N) = \sum_{i=1}^N \sum_{k=1}^K \left[ \mathbb{E}_{\tau \sim \pi_i^k, P_{\mathcal{D}_i}^*, P^*} \left[ c(\tau \,|\, s_1, d_0) \right] - \mathbb{E}_{\tau \sim \pi_i^*, P_{\mathcal{D}_i}^*, P^*} \left[ c(\tau \,|\, s_1, d_0) \right] \right] \tag{15}$$

where $\pi_i^*$ is the optimal switching policy for team $i$, under the true agent policies and environment transition probability.

To achieve our goal, we just run $N$ instances of UCRL2-MC (Algorithm 1), each with a different confidence set $\mathcal{P}_{\mathcal{D}_i}^k(\delta)$ for the agents' policies, similarly as in the case of a single team of agents, but with a shared confidence set $\mathcal{P}^k(\delta)$ for the environment transition probability. Then, we have the following key corollary, which readily follows from Theorem 2:

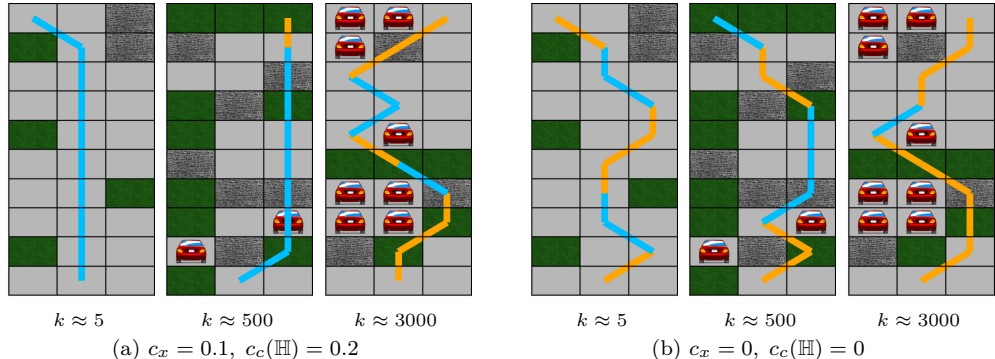

$$k \approx 5 \qquad k \approx 500 \qquad k \approx 3000 \qquad\qquad k \approx 5 \qquad k \approx 500 \qquad k \approx 3000$$

(a) $c_x = 0.1$, $c_c(\mathbb{H}) = 0.2$ $\qquad\qquad$ (b) $c_x = 0$, $c_c(\mathbb{H}) = 0$

Figure 3: Trajectories induced by the switching policies found by Algorithm 1. The blue and orange segments indicate machine and human control, respectively. In both panels, we train Algorithm 1 within the same sequence of episodes, where the initial traffic level of each episode is sampled uniformly from $\{\texttt{no-car}, \texttt{light}, \texttt{heavy}\}$, and show three episodes with different initial traffic levels. The results indicate that, in the latter episodes, the algorithm has learned to switch to the human driver in heavier traffic levels.

**Corollary 3.** *Assume we use $N$ instances of Algorithm 1 to find the switching policies $\pi_i^k$ using a shared confidence set for the environment transition probability. Then, with probability at least $1 - \delta$, it holds that*

$$R(T, N) \leq \rho_1 N L \sqrt{|\mathcal{A}||\mathcal{S}||\mathcal{D}|T \log\left(\frac{|\mathcal{S}||\mathcal{D}|T}{\delta}\right)} + \rho_2 L |\mathcal{S}| \sqrt{|\mathcal{A}|NT \log\left(\frac{|\mathcal{S}||\mathcal{A}|T}{\delta}\right)} \tag{16}$$

*where $\rho_1, \rho_2 > 0$ are constants.*

The above results suggests that our algorithm may achieve lower regret than UCRL2 in a scenario with multiple teams of agents operating in similar environments. This is because, under UCRL2, the confidence sets for the conditional probability defined by Eq. 3 cannot be shared across teams. More specifically, if we use $N$ instances of UCLR2 to find the switching policies $\pi_i^k$, then, with probability at least $1 - \delta$, it holds that

$$R(T, N) \leq \rho N L |\mathcal{S}| \sqrt{|\mathcal{D}|T \log\left(\frac{|\mathcal{S}||\mathcal{D}|T}{\delta}\right)}$$

where $\rho$ is a constant. Then, if we omit constant and logarithmic factors and assume $|\mathcal{D}_i| < |\mathcal{S}|$ for all $i \in [N]$, we have that, for UCRL2, the regret bound is $\tilde{\mathcal{O}}(N L |\mathcal{S}| \sqrt{|\mathcal{D}|T})$ while, for UCRL2-MC, it is $\tilde{\mathcal{O}}(L |\mathcal{S}| \sqrt{|\mathcal{A}|TN} + N L \sqrt{|\mathcal{A}||\mathcal{S}||\mathcal{D}|T})$. Importantly, in practice, we have found that UCRL2-MC does achieve a significant lower regret than UCRL2, as shown in the Figure 5.

## 6 Experiments

### 6.1 Obstacle avoidance

We perform a variety of simulations in obstacle avoidance, where teams of agents (drivers) consist of one human agent ($\mathbb{H}$) and one machine agent ($\mathbb{M}$), i.e., $\mathcal{D} = \{\mathbb{H}, \mathbb{M}\}$. We consider a lane driving environment with three lanes and infinite rows, where the type of each individual cell (*i.e.*, $\texttt{road}$, $\texttt{car}$, $\texttt{stone}$ or $\texttt{grass}$) in row $r$ is sampled independently at random with a probability that depends on the traffic level $\gamma_r$, which can take three discrete values, $\gamma_r \in \{\texttt{no-car}, \texttt{light}, \texttt{heavy}\}$. The traffic level of each row $\gamma_{r+1}$ is sampled at random with a probability that depends on the traffic level of the previous row $\gamma_r$. The probability of each cell type based on traffic level, as well as the conditional distribution of traffic levels can be found in Appendix D.

At any given time $t$, we assume that whoever is in control—be it the machine or the human—can take three different actions $\mathcal{A} = \{\texttt{left}, \texttt{straight}, \texttt{right}\}$. Action $\texttt{left}$ steers the car to the left of the current lane,

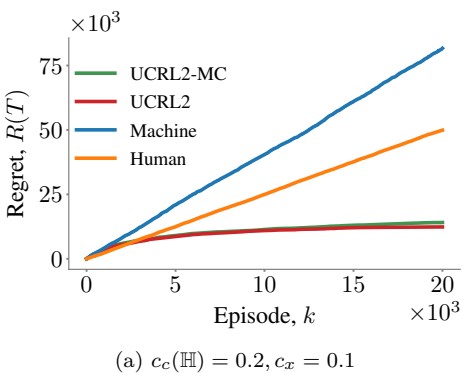 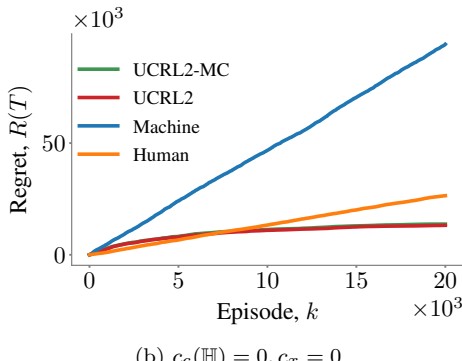

(a) $c_c(\mathbb{H}) = 0.2, c_x = 0.1$

(b) $c_c(\mathbb{H}) = 0, c_x = 0$

Figure 4: Total regret of the trajectories induced by the switching policies found by Algorithm 1 and those induced by a variant of UCRL2 in comparison with the trajectories induced by a machine driver and a human driver in a setting with a single team of agents. In all panels, we run $K = 20,000$. For Algorithm 1 and the variant of UCRL2, the regret is sublinear with respect to the number of time steps whereas, for the machine and the human drivers, the regret is linear.

action `right` steers it to the right and action `straight` leaves the car in the current lane. If the car is already on the leftmost (rightmost) lane when taking action left (right), then the lane remains unchanged. Irrespective of the action taken, the car always moves forward. The goal of the cyberphysical system is to drive the car from an initial state in time $t = 1$ until the end of the episode $t = L$ with the minimum total amount of cost. In our experiments, we set $L = 10$. Figure 2 shows three examples of environment realizations.

**State space.** To evaluate the switching policies found by Algorithm 1, we experiment with a *sensor-based* state space, where the state values are the type of the current cell and the three cells the car can move into in the next time step, as well as the current traffic level—we assume the agents (be it a human or a machine) can measure the traffic level. For example, assume at time $t$ the traffic is light, the car is on a road cell and, if it moves forward left, it hits a stone, if it moves forward straight, it hits a car, and, if it moves forward right, it drives over grass, then its state value is $s_t = (\texttt{light}, \texttt{road}, \texttt{stone}, \texttt{car}, \texttt{grass})$. Moreover, if the car is on the leftmost (rightmost) lane, then we set the value of the third (fifth) dimension in $s_t$ to $\emptyset$. Therefore, under this state representation, the resulting MDP has $\sim 3 \times 5^4$ states.

**Cost and human/machine policies.** We consider a state-dependent environment cost $c_e(s_t, a_t) = c_e(s_t)$ that depends on the type of the cell the car is on at state $s_t$, *i.e.*, $c_e(s_t) = 0$ if the type of the current cell is road, $c_e(s_t) = 2$ if it is grass, $c_e(s_t) = 4$ if it is stone and $c_e(s_t) = 10$ if it is car. Moreover, in all simulations, we use a machine policy that has been trained using a standard RL algorithm on environment realizations with $\gamma_0 = \texttt{no-car}$. In other words, the machine policy is trained to perform well under a low traffic level. Moreover, we consider all the humans pick which action to take (`left`, `straight` or `right`) according to a noisy estimate of the environment cost of the three cells that the car can move into in the next time step. More specifically, each human model $\mathbb{H}$ computes a noisy estimate of the cost $\hat{c}_e(s) = c_e(s) + \epsilon_s$ of each of the three cells the car can move into, where $\epsilon_s \sim N(0, \sigma_{\mathbb{H}})$, and picks the action that moves the car to the cell with the lowest noisy estimate[6]. As a result, human drivers are generally more reliable than the machine under high traffic levels, however, the machine is more reliable than humans under low traffic level, where its policy is near-optimal (See Appendix E for a comparison of the human and machine performance). Finally, we consider that only the car driven by our system moves in the environment.

### 6.1.1 Results

First, we focus on a single team of one machine $\mathbb{M}$ and one human model $\mathbb{H}$, with $\sigma_{\mathbb{H}} = 2$, and use Algorithm 1 to find a sequence of switching policies with sublinear regret. At the beginning of each episode, the initial traffic level $\gamma_0$ is sampled uniformly at random.

---

[6]Note that, in our theoretical results, we have no assumption other than the Markov property regarding the human policy.

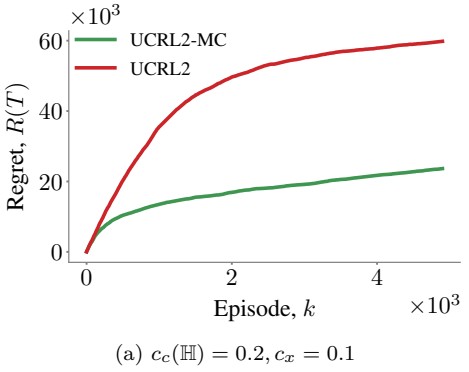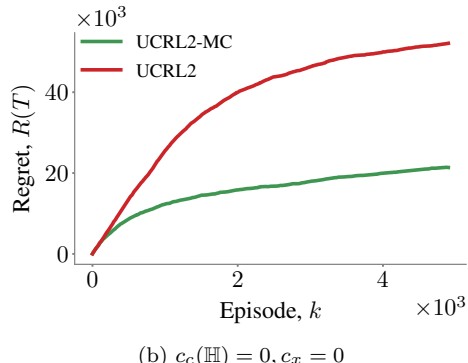

(a) $c_c(\mathbb{H}) = 0.2, c_x = 0.1$                     (b) $c_c(\mathbb{H}) = 0, c_x = 0$

Figure 5: Total regret of the trajectories induced by the switching policies found by $N$ instances of Algorithm 1 and those induced by $N$ instances of a variant of UCRL2 in a setting with $N$ team of agents. In both panels, each instance of Algorithm 1 shares the same confidence set for the environment transition probabilities and we run $K = 5000$ episodes. The sequence of policies found by Algorithm 1 outperform those found by the variant of UCRL2 in terms of total regret, in agreement with Corollary 3.

We look at the trajectories induced by the switching policies found by our algorithm across different episodes for different values of the switching cost $c_x$ and cost of human control $c_c(\mathbb{H})$[7]. Figure 3 summarizes the results, which show that, in the latter episodes, the algorithm has learned to rely on the machine (blue segments) whenever the traffic level is low and switches to the human driver when the traffic level increases. Moreover, whenever the amount of human control and number of switches is not penalized (*i.e.*, $c_x = c_c(\mathbb{H}) = 0$), the algorithm switches to the human more frequently whenever the traffic level is high to reduce the environment cost. See Appendix F for a comparison of human control rate in environments with different initial traffic levels.

In addition, we compare the performance achieved by Algorithm 1 with three baselines: (i) a variant of UCRL2 (Jaksch et al., 2010) adapted to our finite horizon setting (see Appendix C), (ii) a human agent, and (iii) a machine agent. As a measure of performance, we use the total regret, as defined in Eq. 6. Figure 4 summarizes the results for two different values of switching cost $c_x$ and cost of human control $c_c(\mathbb{H})$. The results show that both our algorithm and UCRL2 achieve sublinear regret with respect to the number of time steps and their performance is comparable in agreement with Theorem 2. In contrast, whenever the human or the machine drive on their own, they suffer linear regret, due to a lack of exploration.

Next, we consider $N = 10$ independent teams of agents, $\{\mathcal{D}_i\}_{i=1}^N$, operating in a similar lane driving environment. Each team $\mathcal{D}_i$ is composed of a different human model $\mathbb{H}_i$, with $\sigma_{\mathbb{H}_i}$ sampled uniformly from $(0, 4)$, and the same machine driver $\mathbb{M}$. Then, to find a sequence of switching policies for each of the teams, we run $N$ instances of Algorithm 1 with shared confidence set for the environment transition probabilities.

We compare the performance of our algorithm against the same variant of UCRL2 used in the experiments with a single team of agents in terms of the total regret defined in Eq. 15. Here, note that the variant of UCRL2 does not maintain a shared confidence set for the environment transition probabilities across teams but instead creates a confidence set for the conditional probability defined by Eq. 3 for each team. Figure 5 summarizes the results for a sequence for different values of the switching cost $c_x$ and cost of human control $c_c(\mathbb{H})$, which shows that, in agreement with Corollary 3, our method outperforms UCRL2 significantly.

## 6.2 RiverSwim

In addition to the obstacle avoidance task, we consider the standard task of *RiverSwim* (Strehl & Littman, 2008). The MDP states and transition probabilities are shown in Figure 6. The cost of taking action in states $s_2$ to $s_5$ equals 1, while 0.995 and 0 for states $s_1$ and $s_6$, respectively. Each episode ends after $L = 20$

---

[7]Here, we assume the cost of machine control $c_c(\mathbb{M}) = 0$.

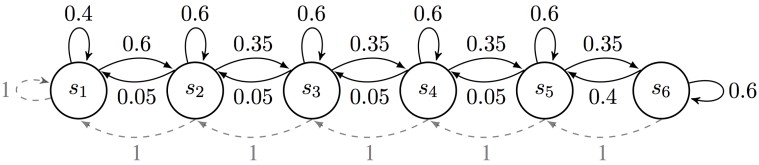

Figure 6: RiverSwim. Continuous (dashed) arrows show the transitions after taking actions `right` (`left`). The optimal policy is to always take action `right`.

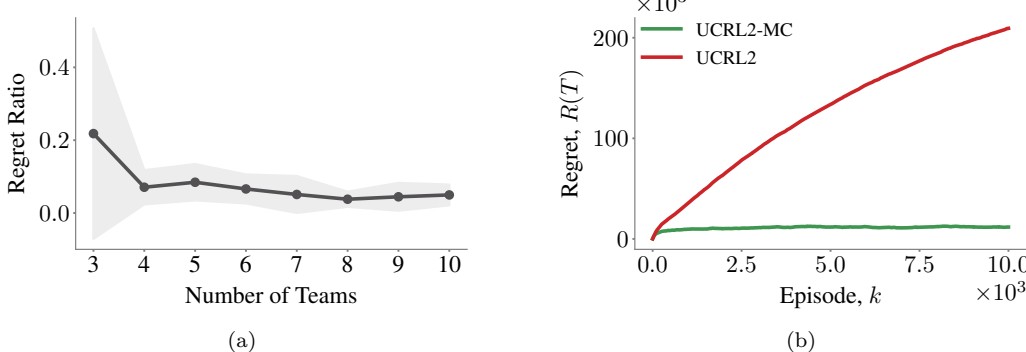

Figure 7: (a) Ratio of UCRL2-MC regret to UCRL2 for different number of teams. (b) Total regret of the trajectories induced by the switching policies found by UCRL2-MC and those induced by UCRL2 in a setting with $N = 100$ team of agents.

steps. We set the switching cost and cost of agent control to zero for all the simulations in this section, i.e., $c_x(\cdot, \cdot) = c_c(\cdot) = 0$. The set $\mathcal{D}$ consists of agents that choose action `right` with some probability value $p$, which may differ for different agents. In the following part, we investigate the effect of increasing the number of teams on the regret bound in the multiple teams of agents setting. See Appendix G for more simulations to study the impact of action size and number of agents in each team on the total regret.

### 6.2.1 Results

We consider $N$ independent teams of agents, each consisting of two agents with the probability $p$ and $1 - p$ of choosing action `right`, where $p$ is chosen uniformly at random for each team. We run the simulations for $N = \{3, 4, \cdots, 10\}$ teams of agents. For each $N$, we run both UCRL2-MC and UCRL2 for 20,000 episodes and repeat each experiment 5 times. Figure 7 (a) summarizes our results, showing the advantage of the shared confidence bounds on the environment transition probabilities in our algorithm against its problem-agnostic version. To better illustrate the performance of UCRL2-MC, we also run an experiment with $N = 100$ teams of agents for 10,000 episodes and compare the total regret of our algorithm to UCRL2. Figure 7 (b) shows that our algorithm significantly outperforms UCRL2.

## 7 Conclusions and Future Work

We have formally defined the problem of learning to switch control among agents in a team via a 2-layer Markov decision process and then developed UCRL2-MC, an online learning algorithm with desirable provable guarantees. Moreover, we have performed a variety of simulation experiments on the standard RiverSwim task and obstacle avoidance to illustrate our theoretical results and demonstrate that, by exploiting the specific structure of the problem, our proposed algorithm is superior to problem-agnostic algorithms.

Our work opens up many interesting avenues for future work. For example, we have assumed that the agents' policies are fixed. However, there are reasons to believe that simultaneously optimizing the agents' policies and the switching policy may lead to superior performance (De et al., 2020; 2021; Wilder et al., 2020; Wu et al., 2020). In our work, we have assumed that the state space is discrete and the horizon in finite. It

would be very interesting to lift these assumptions and develop approximate value iteration methods to solve the learning to switch problem. Finally, it would be interesting to evaluate our algorithm using real human agents in a variety of tasks.

**Acknowledgments.** Gomez-Rodriguez acknowledges support from the European Research Council (ERC) under the European Union's Horizon 2020 research and innovation programme (grant agreement No. 945719).

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

# A    Proofs

## A.1    Proof of Theorem 1

We first define $\mathcal{P}_{\mathcal{D}|\cdot,t^+}^k := \times_{s\in\mathcal{S},d\in\mathcal{D},t'\in\{t,\dots,L\}}\mathcal{P}_{\cdot|d,s,t'}^k$, $\mathcal{P}_{|\cdot,t^+}^k = \times_{s\in\mathcal{S},a\in\mathcal{A},t'\in\{t,\dots,L\}}\mathcal{P}_{|s,a,t'}^k$ and $\pi_{t^+} = \{\pi_t,\dots,\pi_L\}$. Next, we get a lower bound the optimistic value function $v_t^k(s,d)$ as follows:

$$v_t^k(s,d)$$
$$= \min_\pi \min_{P_\mathcal{D}\in\mathcal{P}_\mathcal{D}^k} \min_{P\in\mathcal{P}^k} V_{t|P_\mathcal{D},P}^\pi(s,d)$$
$$= \min_{\pi_{t^+}} \min_{P_\mathcal{D}\in\mathcal{P}_\mathcal{D}^k} \min_{P\in\mathcal{P}^k} V_{t|P_\mathcal{D},P}^\pi(s,d)$$
$$\overset{(i)}{=} \min_{\pi_t(s,d)} \min_{p_{\pi_t(s,d)}(.|s,t)\in\mathcal{P}_{\cdot\,|\,\pi_t(s,d),s,t}^k} \min_{p(.|s,.,t)\in\mathcal{P}_{\cdot\,|\,s,\cdot,t}^k}$$
$$\min_{\substack{\pi_{(t+1)^+}\\P_\mathcal{D}\in\mathcal{P}_{\mathcal{D}\,|\,\cdot,(t+1)^+}^k\\P\in\mathcal{P}_{\cdot\,|\,\cdot,(t+1)^+}^k}} \left[c_{\pi_t(s,d)}(s,d) + \mathbb{E}_{a\sim p_{\pi_t(s,d)}(\cdot\,|\,s,t)}\left(c_e(s,a) + \mathbb{E}_{s'\sim p(\cdot\,|\,s,a,t)}V_{t+1|P_\mathcal{D},P}^\pi(s',\pi_t(s,d))\right)\right]$$
$$\overset{(ii)}{\geq} \min_{\pi_t(s,d)} \min_{p_{\pi_t(s,d)}(.|s,t)\in\mathcal{P}_{\cdot\,|\,\pi_t(s,d),s,t}^k} \min_{p(.|s,.,t)\in\mathcal{P}_{\cdot\,|\,s,\cdot,t}^k} \left[c_{\pi_t(s,d)}(s,d)\right.$$
$$\left.+\mathbb{E}_{a\sim p_{\pi_t(s,d)}(\cdot\,|\,s,t)}\left(c_e(s,a) + \mathbb{E}_{s'\sim p(\cdot\,|\,s,a,t)}\left[\min_{\pi_{(t+1)^+}} \min_{P_\mathcal{D}\in\mathcal{P}_{\mathcal{D}\,|\,\cdot,(t+1)^+}^k} \min_{P\in\mathcal{P}_{\cdot\,|\,\cdot,(t+1)^+}^k} V_{t+1|P_\mathcal{D},P}^\pi(s',\pi_t(s,d))\right]\right)\right]$$
$$= \min_{d_t}\left[c_{d_t}(s,d) + \min_{p_{d_t}(.|s,t)\in\mathcal{P}_{\cdot|d_t,s,t}^k} \sum_{a\in\mathcal{A}} p_{d_t}(a|s,t)\cdot\left(c_e(s,a) + \min_{p(.|s,a,t)\in\mathcal{P}_{\cdot\,|\,s,a,t}^k} \mathbb{E}_{s'\sim p(\cdot\,|\,s,a,t)}v_{t+1}^k(s',d_t)\right)\right],$$

where (i) follows from Lemma 8 and (ii) follows from the fact that $\min_a \mathbb{E}[X(a)] \geq E[\min_a X(a)]$. Next, we provide an upper bound of the optimistic value function $v_t^k(s,d)$ as follows:

$$v_t^k(s,d)$$
$$= \min_\pi \min_{P_\mathcal{D}\in\mathcal{P}_\mathcal{D}^k} \min_{P\in\mathcal{P}^k} V_{t|P_\mathcal{D},P}^\pi(s,d)$$
$$\overset{(i)}{=} \min_{\pi_t} \min_{p_{\pi_t(s,d)}(.|s,t)\in\mathcal{P}_{\cdot\,|\,\pi_t(s,d),s,t}^k} \min_{p(.|s,.,t)\in\mathcal{P}_{\cdot\,|\,s,\cdot,t}^k}$$
$$\min_{\substack{\pi_{(t+1)^+}\\P_\mathcal{D}\in\mathcal{P}_{\mathcal{D}\,|\,\cdot,(t+1)^+}^k\\P\in\mathcal{P}_{\cdot\,|\,\cdot,(t+1)^+}^k}} \left[c_{\pi_t(s,d)}(s,d) + \mathbb{E}_{a\sim p_{\pi_t(s,d)}(\cdot\,|\,s,t)}\left(c_e(s,a) + \mathbb{E}_{s'\sim p(\cdot\,|\,s,a,t)}V_{t+1|P_\mathcal{D},P}^\pi(s',\pi_t(s,d))\right)\right]$$
$$\overset{(ii)}{\leq} \min_{\pi_t(s,d)} \min_{p_{\pi_t(s,d)}(.|s,t)\in\mathcal{P}_{\cdot\,|\,\pi_t(s,d),s,t}^k}$$
$$\min_{p(.|s,.,t)\in\mathcal{P}_{\cdot\,|\,s,\cdot,t}^k} \left[c_{\pi_t(s,d)}(s,d) +\mathbb{E}_{a\sim p_{\pi_t(s,d)}(\cdot\,|\,s,t)}\left(c_e(s,a) + \mathbb{E}_{s'\sim p(\cdot\,|\,s,a,t)}V_{t+1|P_\mathcal{D}^*,P^*}^{\pi^*}(s',\pi_t(s,d))\right)\right]$$
$$\overset{(iii)}{=} \min_{\pi_t(s,d)} \min_{p_{\pi_t(s,d)}(.|s,t)\in\mathcal{P}_{\cdot\,|\,\pi_t(s,d),s,t}^k}$$
$$\min_{p(.|s,.,t)\in\mathcal{P}_{\cdot\,|\,s,\cdot,t}^k} \left[c_{\pi_t(s,d)}(s,d) +\mathbb{E}_{a\sim p_{\pi_t(s,d)}(\cdot\,|\,s,t)}\left(c_e(s,a) + \mathbb{E}_{s'\sim p(\cdot\,|\,s,a,t)}v_{t+1}^k(s',\pi_t(s,d))\right)\right]$$
$$= \min_{d_t}\left[c_{d_t}(s,d) + \min_{p_{d_t}(.|s,t)\in\mathcal{P}_{\cdot|d_t,s,t}^k} \sum_{a\in\mathcal{A}} p_{d_t}(a|s,t)\cdot\left(c_e(s,a) + \min_{p(.|s,a,t)\in\mathcal{P}_{\cdot\,|\,s,a,t}^k} \mathbb{E}_{s'\sim p(\cdot\,|\,s,a,t)}v_{t+1}^k(s',d_t)\right)\right].$$

Here, (i) follows from Lemma 8, (ii) follows from the fact that:

$$\min_{\substack{\pi_{(t+1)+} \\ P_{\mathcal{D}} \in \mathcal{P}^k_{\mathcal{D}\,|\,\cdot,(t+1)+} \\ P \in \mathcal{P}^k_{\cdot\,|\,\cdot,(t+1)+}}} \left[ c_{\pi_t(s,d)}(s,d) + \mathbb{E}_{a \sim p_{\pi_t(s,d)}(\cdot\,|\,s,t)} \left( c_e(s,a) + \mathbb{E}_{s' \sim p(\cdot\,|\,s,a,t)} V^{\pi}_{t+1|P_{\mathcal{D}},P}(s',\pi_t(s,d)) \right) \right]$$

$$\leq \left[ c_{\pi_t(s,d)}(s,d) + \mathbb{E}_{a \sim p_{\pi_t(s,d)}(\cdot\,|\,s,t)} \left( c_e(s,a) + \mathbb{E}_{s' \sim p(\cdot\,|\,s,a,t)} V^{\pi}_{t+1|P_{\mathcal{D}},P}(s',\pi_t(s,d)) \right) \right]$$

$$\forall \pi, P_{\mathcal{D}} \in \mathcal{P}^k_{\mathcal{D}\,|\,\cdot,(t+1)+}, P \in \mathcal{P}^k_{\cdot\,|\,\cdot,(t+1)+} \tag{17}$$

and if we set $\pi_{(t+1)+} = \{\pi^*_{t+1}, ..., \pi^*_L\}$, $P_{\mathcal{D}} = P^*_{\mathcal{D}} \in \mathcal{P}^k_{\mathcal{D}\,|\,\cdot,(t+1)+}$ and $P = P^* \in \mathcal{P}^k_{\cdot\,|\,\cdot,(t+1)+}$, where

$$\{\pi^*_{t+1}, ..., \pi^*_L\}, P^*_{\mathcal{D}}, P^* = \underset{\substack{\pi_{(t+1)+} \\ P_{\mathcal{D}} \in \mathcal{P}^k_{\mathcal{D}\,|\,\cdot,(t+1)+} \\ P \in \mathcal{P}^k_{\cdot\,|\,\cdot,(t+1)+}}}{\operatorname{argmin}} V^{\pi}_{t+1|P_{\mathcal{D}},P}(s',\pi_t(s,d)), \tag{18}$$

then equality (iii) holds. Since the upper and lower bounds are the same, we can conclude that the optimistic value function satisfies Eq. 12, which completes the proof.

## A.2 Proof of Theorem 2

In this proof, we assume that $c_e(s,a) + c_c(d) + c_x(d,d') < 1$ for all $s \in \mathcal{S}, a \in \mathcal{A}$ and $d, d' \in \mathcal{D}$. Throughout the proof, we will omit the subscripts $P^*_{\mathcal{D}}, P^*$ in $V_{t\,|\,P^*_{\mathcal{D}},P^*}$ and write $V_t$ instead in case of true agent policies $P^*_{\mathcal{D}}$ and true transition probabilities $P^*$. Then, we define the following quantities:

$$P^k_{\mathcal{D}} = \underset{P_{\mathcal{D}} \in \mathcal{P}^k_{\mathcal{D}}(\delta)}{\operatorname{argmin}} \min_{P \in \mathcal{P}^k(\delta)} V^{\pi^k}_{1|P_{\mathcal{D}},P}(s_1,d_0), \tag{19}$$

$$P^k = \underset{P \in \mathcal{P}^k(\delta)}{\operatorname{argmin}} V^{\pi^k}_{1|P^k_{\mathcal{D}},P}(s_1,d_0), \tag{20}$$

$$\Delta_k = V^{\pi^k}_1(s_1,d_0) - V^{\pi^*}_1(s_1,d_0), \tag{21}$$

where, recall from Eq. 7 that, $\pi^k = \operatorname{argmin}_\pi \min_{P_{\mathcal{D}} \in \mathcal{P}^k_{\mathcal{D}}}, \min_{P \in \mathcal{P}^k} V^{\pi}_{1|P_{\mathcal{D}},P}(s_1,d_0)$; and, $\Delta_k$ indicates the regret for episode $k$. Hence, we have

$$R(T) = \sum_{k=1}^K \Delta_k = \sum_{k=1}^K \Delta_k \mathbb{I}(P^*_{\mathcal{D}} \in \mathcal{P}^k_{\mathcal{D}} \wedge P^* \in \mathcal{P}^k) + \sum_{k=1}^K \Delta_k \mathbb{I}(P^*_{\mathcal{D}} \notin \mathcal{P}^k_{\mathcal{D}} \vee P^* \notin \mathcal{P}^k) \tag{22}$$

Next, we split the analysis into two parts. We first bound $\sum_{k=1}^K \Delta_k \mathbb{I}(P^*_{\mathcal{D}} \in \mathcal{P}^k_{\mathcal{D}} \wedge P^* \in \mathcal{P}^k)$ and then bound $\sum_{k=1}^K \Delta_k \mathbb{I}(P^*_{\mathcal{D}} \notin \mathcal{P}^k_{\mathcal{D}} \vee P^* \notin \mathcal{P}^k)$.

— *Computing the bound on* $\sum_{k=1}^K \Delta_k \mathbb{I}(P^*_{\mathcal{D}} \in \mathcal{P}^k_{\mathcal{D}} \wedge P^* \in \mathcal{P}^k)$

First, we note that

$$\Delta_k = V^{\pi^k}_1(s_1,d_0) - V^{\pi^*}_1(s_1,d_0) \leq V^{\pi^k}_1(s_1,d_0) - V^{\pi^k}_{1|P^k_{\mathcal{D}},P^k}(s_1,d_0) \tag{23}$$

This is because

$$V^{\pi^k}_{1|P^k_{\mathcal{D}},P^k}(s_1,d_0) \overset{(i)}{=} \min_\pi \min_{P_{\mathcal{D}} \in \mathcal{P}^k_{\mathcal{D}}} \min_{P \in \mathcal{P}^k} V^{\pi}_{1|P_{\mathcal{D}},P}(s_1,d_0) \overset{(ii)}{\leq} \min_\pi V^{\pi}_{1|P^*_{\mathcal{D}},P^*}(s_1,d_0) = V^{\pi^*}_1(s_1,d_0), \tag{24}$$

where (i) follows from Eqs. 19, 20, and (ii) holds because of the fact that the true transition probabilities $P^*_{\mathcal{D}} \in \mathcal{P}^k_{\mathcal{D}}$ and $P^* \in \mathcal{P}^k$. Next, we use Lemma 4 (Appendix B) to bound $\sum_{k=1}^K (V^{\pi^k}_1(s_1,d_0) - V^{\pi^k}_{1|P^k_{\mathcal{D}},P^k}(s_1,d_0))$.

$$\sum_{k=1}^K (V^{\pi^k}_1(s_1,d_0) - V^{\pi^k}_{1|P^k_{\mathcal{D}},P^k}(s_1,d_0)) \leq \sum_{k=1}^K L\mathbb{E}\left[ \sum_{t=1}^L \min\{1, \beta^k_{\mathcal{D}}(s_t,d_t,\delta)\} + \sum_{t=1}^L \min\{1, \beta^k(s_t,a_t\delta)\} \Big| s_1,d_0 \right] \tag{25}$$

Since by assumption, $c_e(s,a) + c_c(d) + c_x(d,d') < 1$ for all $s \in \mathcal{S}, a \in \mathcal{A}$ and $d, d' \in \mathcal{D}$, the worst-case regret is bounded by $T$. Therefore, we have that:

$$
\begin{aligned}
\sum_{k=1}^{K} \Delta_k \mathbb{I}(P_{\mathcal{D}}^* \in \mathcal{P}_{\mathcal{D}}^k \wedge P^* \in \mathcal{P}^k) &\leq \min \left\{ T, \sum_{k=1}^{K} L\mathbb{E}\left[ \sum_{t=1}^{L} \min\{1, \beta_{\mathcal{D}}^k(s_t, d_t, \delta)\} | s_1, d_0 \right] \right. \\
&\qquad\qquad \left. + \sum_{k=1}^{K} L\mathbb{E}\left[ \sum_{t=1}^{L} \min\{1, \beta^k(s_t, a_t, \delta)\} | s_1, d_0 \right] \right\} \\
&\leq \min \left\{ T, \sum_{k=1}^{K} L\mathbb{E}\left[ \sum_{t=1}^{L} \min\{1, \beta_{\mathcal{D}}^k(s_t, d_t, \delta)\} | s_1, d_0 \right] \right\} \\
&\qquad\qquad + \min \left\{ T, \sum_{k=1}^{K} L\mathbb{E}\left[ \sum_{t=1}^{L} \min\{1, \beta^k(s_t, a_t, \delta)\} | s_1, d_0 \right] \right\}, \quad (26)
\end{aligned}
$$

where, the last inequality follows from Lemma 9. Now, we aim to bound the first term in the RHS of the above inequality.

$$
\begin{aligned}
\sum_{k=1}^{K} L\mathbb{E}\left[ \sum_{t=1}^{L} \min\{1, \beta_{\mathcal{D}}^k(s_t, d_t, \delta)\} | s_1, d_0 \right] &\overset{(i)}{=} L \sum_{k=1}^{K} \mathbb{E}\left[ \sum_{t=1}^{L} \min\left\{1, \sqrt{\frac{2\log\left(\frac{((k-1)L)^7 |\mathcal{S}||\mathcal{D}|2^{|\mathcal{A}|+1}}{\delta}\right)}{\max\{1, N_k(s_t, d_t)\}}}\right\} \Big| s_1, d_0 \right] \\
&\overset{(ii)}{\leq} L \sum_{k=1}^{K} \mathbb{E}\left[ \sum_{t=1}^{L} \min\left\{1, \sqrt{\frac{2\log\left(\frac{(KL)^7 |\mathcal{S}||\mathcal{D}|2^{|\mathcal{A}|+1}}{\delta}\right)}{\max\{1, N_k(s_t, d_t)\}}}\right\} \right] \\
&\overset{(iii)}{\leq} 2\sqrt{2}L\sqrt{2\log\left(\frac{(KL)^7 |\mathcal{S}||\mathcal{D}|2^{|\mathcal{A}|+1}}{\delta}\right) |\mathcal{S}||\mathcal{D}|KL} \\
&\qquad + 2L^2|\mathcal{S}||\mathcal{D}| \quad\quad (27) \\
&\leq 2\sqrt{2}\sqrt{14|\mathcal{A}|\log\left(\frac{(KL)|\mathcal{S}||\mathcal{D}|}{\delta}\right) |\mathcal{S}||\mathcal{D}|KL} + 2L^2|\mathcal{S}||\mathcal{D}| \\
&= \sqrt{112}\sqrt{|\mathcal{A}|\log\left(\frac{(KL)|\mathcal{S}||\mathcal{D}|}{\delta}\right) |\mathcal{S}||\mathcal{D}|KL} + 2L^2|\mathcal{S}||\mathcal{D}| \quad (28)
\end{aligned}
$$

where (i) follows by replacing $\beta_{\mathcal{D}}^k(s_t, d_t, \delta)$ with its definition, (ii) follows by the fact that $(k-1)L \leq KL$, (iii) follows from Lemma 5, in which, we put $\mathcal{W} := \mathcal{S} \times \mathcal{D}$, $c := \sqrt{2\log\left(\frac{(KL)^7 |\mathcal{S}||\mathcal{D}|2^{|\mathcal{A}|+1}}{\delta}\right)}$, $\mathcal{T}_k = (w_{k,1}, \ldots, w_{k,L}) := ((s_1, d_1), \ldots, (s_L, d_L))$. Now, due to Eq. 28, we have the following.

$$
\min\left\{ T, \sum_{k=1}^{K} L\mathbb{E}\left[ \sum_{t=1}^{L} \min\{1, \beta_{\mathcal{D}}^k(s_t, d_t, \delta)\} | s_1, d_0 \right] \right\} \leq \min\left\{ T, \sqrt{112}L\sqrt{|\mathcal{A}||\mathcal{S}||\mathcal{D}|T\log\left(\frac{T|\mathcal{S}||\mathcal{D}|}{\delta}\right)} + 2L^2|\mathcal{S}||\mathcal{D}| \right\} \quad (29)
$$

Now, if $T \leq 2L^2|\mathcal{S}||\mathcal{A}||\mathcal{D}|\log\left(\frac{T|\mathcal{S}||\mathcal{D}|}{\delta}\right)$,

$$
T^2 \leq 2L^2|\mathcal{S}||\mathcal{A}||\mathcal{D}|T\log\left(\frac{T|\mathcal{S}||\mathcal{D}|}{\delta}\right) \implies T \leq \sqrt{2}L\sqrt{|\mathcal{S}||\mathcal{A}||\mathcal{D}|T\log\left(\frac{T|\mathcal{S}||\mathcal{D}|}{\delta}\right)}
$$

and if $T > 2L^2 |\mathcal{S}||\mathcal{A}||\mathcal{D}| \log \left( \frac{T|\mathcal{S}||\mathcal{D}|}{\delta} \right)$,

$$2L^2|\mathcal{S}| < \frac{\sqrt{2L^2|\mathcal{S}||\mathcal{A}||\mathcal{D}|T \log \left( \frac{T|\mathcal{S}||\mathcal{D}|}{\delta} \right)}}{|\mathcal{A}||\mathcal{D}|\log \left( \frac{T|\mathcal{S}||\mathcal{D}|}{\delta} \right)} \leq \sqrt{2}L \sqrt{|\mathcal{S}||\mathcal{A}||\mathcal{D}|T \log \left( \frac{T|\mathcal{S}||\mathcal{D}|}{\delta} \right)}. \tag{30}$$

Thus, the minimum in Eq. 29 is less than

$$(\sqrt{2} + \sqrt{112})L\sqrt{|\mathcal{S}||\mathcal{A}||\mathcal{D}|T \log \left( \frac{|\mathcal{S}||\mathcal{D}|T}{\delta} \right)} < 12L\sqrt{|\mathcal{S}||\mathcal{A}||\mathcal{D}|T \log \left( \frac{|\mathcal{S}||\mathcal{D}|T}{\delta} \right)} \tag{31}$$

A similar analysis can be done for the second term of the RHS of Eq. 26, which would show that,

$$\min \left\{ T, \sum_{k=1}^{K} L\mathbb{E} \left[ \sum_{t=1}^{L} \min\{1, \beta^k(s_t, a_t, \delta)\} | s_1, d_0 \right] \right\} \leq 12L|\mathcal{S}|\sqrt{|\mathcal{A}|T \log \left( \frac{T|\mathcal{S}||\mathcal{A}|}{\delta} \right)}. \tag{32}$$

Combining Eqs. 26, 31 and 32, we can bound the first term of the total regret as follows:

$$\sum_{k=1}^{K} \Delta_k \mathbb{I}(P_{\mathcal{D}}^* \in \mathcal{P}_{\mathcal{D}}^k \wedge P^* \in \mathcal{P}^k) \leq 12L\sqrt{|\mathcal{A}||\mathcal{S}||\mathcal{D}|T \log \left( \frac{T|\mathcal{S}||\mathcal{D}|}{\delta} \right)} + 12L|\mathcal{S}|\sqrt{|\mathcal{A}|T \log \left( \frac{T|\mathcal{S}||\mathcal{A}|}{\delta} \right)} \tag{33}$$

— *Computing the bound on $\sum_{k=1}^{K} \Delta_k \mathbb{I}(P_{\mathcal{D}}^* \notin \mathcal{P}_{\mathcal{D}}^k \vee P^* \notin \mathcal{P}^k)$*

Here, we use a similar approach to Jaksch et al. (2010). Note that

$$\sum_{k=1}^{K} \Delta_k \mathbb{I}(P_{\mathcal{D}}^* \notin \mathcal{P}_{\mathcal{D}}^k \vee P^* \notin \mathcal{P}^k) = \sum_{k=1}^{\left\lfloor \sqrt{\frac{K}{L}} \right\rfloor} \Delta_k \mathbb{I}(P_{\mathcal{D}}^* \notin \mathcal{P}_{\mathcal{D}}^k \vee P^* \notin \mathcal{P}^k) + \sum_{k=\left\lfloor \sqrt{\frac{K}{L}} \right\rfloor + 1}^{K} \Delta_k \mathbb{I}(P_{\mathcal{D}}^* \notin \mathcal{P}_{\mathcal{D}}^k \vee P^* \notin \mathcal{P}^k). \tag{34}$$

Now, our goal is to show the second term of the RHS of above equation vanishes with high probability. If we succeed, then it holds that, with high probability, $\sum_{k=1}^{K} \Delta_k \mathbb{I}(P_{\mathcal{D}}^* \notin \mathcal{P}_{\mathcal{D}}^k \vee P^* \notin \mathcal{P}^k)$ equals the first term of the RHS and then we will be done because

$$\sum_{k=1}^{\left\lfloor \sqrt{\frac{K}{L}} \right\rfloor} \Delta_k \mathbb{I}(P_{\mathcal{D}}^* \notin \mathcal{P}_{\mathcal{D}}^k \vee P^* \notin \mathcal{P}^k) \leq \sum_{k=1}^{\left\lfloor \sqrt{\frac{K}{L}} \right\rfloor} \Delta_k \overset{(i)}{\leq} \left\lfloor \sqrt{\frac{K}{L}} \right\rfloor L = \sqrt{KL}, \tag{35}$$

where (i) follows from the fact that $\Delta_k \leq L$ since we assumed the cost of each step $c_e(s,a) + c_c(d) + c_x(d,d') \leq 1$ for all $s \in \mathcal{S}$, $a \in \mathcal{A}$, and $d, d' \in \mathcal{D}$.

To prove that $\sum_{k=\left\lfloor \sqrt{\frac{K}{L}} \right\rfloor + 1}^{K} \Delta_k \mathbb{I}(P_{\mathcal{D}}^* \notin \mathcal{P}_{\mathcal{D}}^k \vee P^* \notin \mathcal{P}^k) = 0$ with high probability, we proceed as follows. By applying Lemma 6 to $P_{\mathcal{D}}^*$ and $P^*$, we have

$$\Pr(P_{\mathcal{D}}^* \notin \mathcal{P}_{\mathcal{D}}^k) \leq \frac{\delta}{2t_k^6}, \ \Pr(P^* \notin \mathcal{P}^k) \leq \frac{\delta}{2t_k^6} \tag{36}$$

Thus,

$$\Pr(P_{\mathcal{D}}^* \notin \mathcal{P}_{\mathcal{D}}^k \vee P^* \notin \mathcal{P}^k) \leq \Pr(P_{\mathcal{D}}^* \notin \mathcal{P}_{\mathcal{D}}^k) + \Pr(P^* \notin \mathcal{P}^k) \leq \frac{\delta}{t_k^6} \tag{37}$$

where $t_k = (k-1)L$ is the end time of episode $k - 1$. Therefore, it follows that

$$\Pr\left(\sum_{k=\lfloor\sqrt{\frac{K}{L}}\rfloor+1}^{K} \Delta_k \mathbb{I}(P_{\mathcal{D}}^* \notin \mathcal{P}_{\mathcal{D}}^k \vee P^* \notin \mathcal{P}^k) = 0\right) = \Pr\left(\forall k : \left\lfloor\sqrt{\frac{K}{L}}\right\rfloor + 1 \leq k \leq K; \ P_{\mathcal{D}}^* \in \mathcal{P}_{\mathcal{D}}^k \wedge P^* \in \mathcal{P}^k\right)$$

$$= 1 - \Pr\left(\exists k : \left\lfloor\sqrt{\frac{K}{L}}\right\rfloor + 1 \leq k \leq K; \ P_{\mathcal{D}}^* \notin \mathcal{P}_{\mathcal{D}}^k \vee P^* \notin \mathcal{P}^k\right)$$

$$\overset{(i)}{\geq} 1 - \sum_{k=\lfloor\sqrt{\frac{K}{L}}\rfloor+1}^{K} \Pr(P_{\mathcal{D}}^* \notin \mathcal{P}_{\mathcal{D}}^k \vee P^* \notin \mathcal{P}^k)$$

$$\overset{(ii)}{\geq} 1 - \sum_{k=\lfloor\sqrt{\frac{K}{L}}\rfloor+1}^{K} \frac{\delta}{t_k^6}$$

$$\overset{(iii)}{\geq} 1 - \sum_{t=\sqrt{KL}}^{KL} \frac{\delta}{t^6} \geq 1 - \int_{\sqrt{KL}}^{KL} \frac{\delta}{t^6} \geq 1 - \frac{\delta}{5(KL)^{\frac{5}{4}}}. \tag{38}$$

where (i) follows from a union bound, (ii) follows from Eq. 37 and (iii) holds using that $t_k = (k-1)L$. Hence, with probability at least $1 - \frac{\delta}{5(KL)^{\frac{5}{4}}}$ we have that

$$\sum_{k=\lfloor\sqrt{\frac{K}{L}}\rfloor+1}^{K} \Delta_k \mathbb{I}(P_{\mathcal{D}}^* \notin \mathcal{P}_{\mathcal{D}}^k \vee P^* \notin \mathcal{P}^k) = 0. \tag{39}$$

If we combine the above equation and Eq. 35, we can conclude that, with probability at least $1 - \frac{\delta}{5T^{5/4}}$, we have that

$$\sum_{k=1}^{\lfloor\sqrt{\frac{K}{L}}\rfloor} \Delta_k \mathbb{I}(P_{\mathcal{D}}^* \notin \mathcal{P}_{\mathcal{D}}^k \vee P^* \notin \mathcal{P}^k) \leq \sqrt{T} \tag{40}$$

where $T = KL$. Next, if we combine Eqs. 33 and 40, we have

$$R(T) = \sum_{k=1}^{K} \Delta_k \mathbb{I}(P_{\mathcal{D}}^* \in \mathcal{P}_{\mathcal{D}}^k \wedge P^* \in \mathcal{P}^k) + \sum_{k=1}^{K} \Delta_k \mathbb{I}(P_{\mathcal{D}}^* \notin \mathcal{P}_{\mathcal{D}}^k \vee P^* \notin \mathcal{P}^k)$$

$$\leq 12L\sqrt{|\mathcal{A}||\mathcal{S}||\mathcal{D}|T \log\left(\frac{T|\mathcal{S}||\mathcal{D}|}{\delta}\right)} + 12L|\mathcal{S}|\sqrt{|\mathcal{A}|T \log\left(\frac{T|\mathcal{S}||\mathcal{A}|}{\delta}\right)} + \sqrt{T}$$

$$\leq 13L\sqrt{|\mathcal{A}||\mathcal{S}||\mathcal{D}|T \log\left(\frac{T|\mathcal{S}||\mathcal{D}|}{\delta}\right)} + 12L|\mathcal{S}|\sqrt{|\mathcal{A}|T \log\left(\frac{T|\mathcal{S}||\mathcal{A}|}{\delta}\right)} \tag{41}$$

Finally, since $\sum_{T=1}^{\infty} \frac{\delta}{5T^{5/4}} \leq \delta$, with probability at least $1 - \delta$, the above inequality holds. This concludes the proof.

## B Useful lemmas

**Lemma 4.** *Suppose $P_{\mathcal{D}}$ and $P$ are true transitions and $P_{\mathcal{D}} \in \mathcal{P}_{\mathcal{D}}^k$, $P \in \mathcal{P}^k$ for episode $k$. Then, for arbitrary policy $\pi^k$, and arbitrary $P_{\mathcal{D}}^k \in \mathcal{P}_{\mathcal{D}}^k$, $P^k \in \mathcal{P}^k$, it holds that*

$$V_{1|P_{\mathcal{D}},P}^{\pi^k}(s,d) - V_{1|P_{\mathcal{D}}^k,P^k}^{\pi^k}(s,d) \leq L\mathbb{E}\left[\sum_{t=1}^{L} \min\{1, \beta_{\mathcal{D}}^k(s_t, d_t, \delta)\} + \sum_{t=1}^{H} \min\{1, \beta^k(s_t, a_t, \delta)\} \mid s_1 = s, d_0 = d\right], \tag{42}$$

*where the expectation is taken over the MDP with policy $\pi^k$ under true transitions $P_{\mathcal{D}}$ and $P$.*

*Proof.* For ease of notation, let $\overline{v}_t^k := V_{t\,|\,P_{\mathcal{D}},P}^{\pi^k}$, $\overline{v}_{t\,|\,k}^k := V_{t\,|\,P_{\mathcal{D}}^k,P^k}^{\pi^k}$ and $c_t^{\pi}(s,d) = c_{\pi_t^k(s,d)}(s,d)$. We also define $d' = \pi_1^k(s,d)$. From Eq 68, we have

$$\overline{v}_1^k(s,d) = c_1^{\pi}(s,d) + \sum_{a\in\mathcal{A}} p_{\pi_1^k(s,d)}(a\,|\,s)\cdot\left(c_e(s,a) + \sum_{s'\in\mathcal{S}} p(s'|s,a)\cdot\overline{v}_2^k(s',d')\right) \tag{43}$$

$$\overline{v}_{1\,|\,k}^k(s,d) = c_1^{\pi}(s,d) + \sum_{a\in\mathcal{A}} p_{\pi_1^k(s,d)}^k(a\,|\,s)\cdot\left(c_e(s,a) + \sum_{s'\in\mathcal{S}} p^k(s'|s,a)\cdot\overline{v}_{2\,|\,k}^k(s',d')\right) \tag{44}$$

Now, using above equations, we rewrite $\overline{v}_1^k(s,d) - \overline{v}_{1\,|\,k}^k(s,d)$ as

$$\overline{v}_1^k(s,d) - \overline{v}_{1\,|\,k}^k(s,d) = \sum_{a\in\mathcal{A}} p_{\pi_1^k(s,d)}(a|s)\left(c_e(s,a) + \sum_{s'\in\mathcal{S}} p(s'|s,a)\cdot\overline{v}_2^k(s',d')\right)$$

$$- \sum_{a\in\mathcal{A}} p_{\pi_1^k(s,d)}^k(a|s)\left(c_e(s,a) + \sum_{s'\in\mathcal{S}} p^k(s'|s,a)\cdot\overline{v}_{2\,|\,k}^k(s',d')\right)$$

$$\overset{(i)}{=} \sum_{a\in\mathcal{A}} p_{\pi_1^k(s,d)}(a\,|\,s)\left(c_e(s,a) + \sum_{s'\in\mathcal{S}} p(s'|s,a)\cdot\overline{v}_2^k(s',d') - c_e(s,a) - \sum_{s'\in\mathcal{S}} p^k(s'\,|\,s)\cdot\overline{v}_{2\,|\,k}^k(s',d')\right)$$

$$+ \sum_{a\in\mathcal{A}}\left(p_{\pi_1^k(s,d)}(a\,|\,s) - p_{\pi_1^k(s,d)}^k(a\,|\,s)\right)\left(\underbrace{c_e(s,a) + \sum_{s'\in\mathcal{S}} p^k(s'\,|\,s,a)\cdot\overline{v}_{2\,|\,k}^k(s',d')}_{\leq L}\right)$$

$$\overset{(ii)}{\leq} \sum_{a\in\mathcal{A}}\left[p_{\pi_1^k(s,d)}(a\,|\,s)\cdot\sum_{s'\in\mathcal{S}}\left[p(s'\,|\,s,a)\overline{v}_2^k(s',d') - p^k(s'\,|\,s,a)\overline{v}_{2\,|\,k}^k(s',d')\right]\right]$$

$$+ L\sum_{a\in\mathcal{A}}\left[p_{\pi_1^k(s,d)}(a\,|\,s) - p_{\pi_1^k(s,d)}^k(a\,|\,s)\right]$$

$$\overset{(iii)}{=} \sum_{a\in\mathcal{A}}\left[p_{\pi_1^k(s,d)}(a\,|\,s)\cdot\sum_{s'\in\mathcal{S}} p(s'\,|\,s,a)\cdot\left(\overline{v}_2^k(s',d') - \overline{v}_{2\,|\,k}^k(s',d')\right)\right]$$

$$+ \sum_{a\in\mathcal{A}}\left[p_{\pi_1^k(s,d)}(a\,|\,s)\sum_{s'\in\mathcal{S}}\left(p(s'\,|\,s,a) - p^k(s'\,|\,s,a)\right)\underbrace{\overline{v}_{2\,|\,k}^k(s',d')}_{\leq L}\right]$$

$$+ L\sum_{a\in\mathcal{A}}\left[p_{\pi_1^k(s,d)}(a\,|\,s) - p_{\pi_1^k(s,d)}^k(a\,|\,s,d)\right]$$

$$\overset{(iv)}{\leq} \mathbb{E}_{a\sim p_{\pi_1^k(s,d)}(\cdot\,|\,s),s'\sim p(\cdot\,|\,s,a)}\left[\overline{v}_2^k(s',d') - \overline{v}_{2\,|\,k}^k(s',d')\right]$$

$$+ L\mathbb{E}_{a\sim p_{\pi_1^k(s,d)}(\cdot\,|\,s)}\left[\sum_{s'\in\mathcal{S}}\left[p(s'\,|\,s,a) - p^k(s'\,|\,s,a)\right]\right] + L\sum_{a\in\mathcal{A}}\left[p_{\pi_1^k(s,d)}(a\,|\,s) - p_{\pi_1^k(s,d)}^k(a\,|\,s)\right], \tag{45}$$

where (i) follows by adding and subtracting term $p_{\pi_1^k(s,d)}(a\,|\,s)\left(c_e(s,a) + \sum_{s'\in\mathcal{S}} p^k(s'\,|\,s,a)\cdot\overline{v}_{2\,|\,k}^k(s',d')\right)$, (ii) follows from the fact that $c_e(s,a) + \sum_{s'\in\mathcal{S}} p^k(s'\,|\,s,a)\cdot\overline{v}_{2\,|\,k}^k(s',d') \leq L$, since, by assumption, $c_e(s,a) + c_c(d) + c_x(d,d') < 1$ for all $s\in\mathcal{S}, a\in\mathcal{A}$ and $d,d'\in\mathcal{D}$.. Similarly, (iii) follows by adding and subtracting $p(s'\,|\,s,a)\overline{v}_{2\,|\,k}^k(s',d')$, and (iv) follows from the fact that $\overline{v}_{2\,|\,k}^k \leq L$. By assumption, both $P_{\mathcal{D}}$ and $P_{\mathcal{D}}^k$ lie in

the confidence set $\mathcal{P}_{\mathcal{D}}^k(\delta)$, so

$$\sum_{a\in\mathcal{A}}\left[p_{\pi_1^k(s,d)}(a\mid s)-p_{\pi_1^k(s,d)}^k(a\mid s)\right]\leq\min\{1,\beta_{\mathcal{D}}^k(s,d'=\pi_1^k(s,d),\delta)\} \tag{46}$$

Similarly,

$$\sum_{s'\in\mathcal{S}}\left[p(s'\mid s,a)-p^k(s'\mid s,a)\right]\leq\min\{1,\beta^k(s,a,\delta)\} \tag{47}$$

If we combine Eq. 46 and Eq. 47 in Eq. 45, for all $s\in\mathcal{S}$, it holds that

$$\begin{aligned}
\overline{v}_1^k(s,d)-\overline{v}_{1\mid k}^k(s,d)\leq{}&\mathbb{E}_{a\sim p_{\pi_1^k(s,d)}(\cdot|s),s'\sim p(\cdot|s,a)}\left[\overline{v}_2^k(s',d')-\overline{v}_{2\mid k}^k(s',d')\right]\\
&+L\mathbb{E}_{a\sim p_{\pi_1^k(s,d)}(\cdot|s)}\left[\min\{1,\beta^k(s,a,\delta)\}\right]\\
&+L\left[\min\{1,\beta_{\mathcal{D}}^k(s,d'=\pi_1^k(s,d),\delta)\}\right]
\end{aligned} \tag{48}$$

Similarly, for all $s\in\mathcal{S}$, $d\in\mathcal{D}$ we can show

$$\begin{aligned}
\overline{v}_2^k(s,d)-\overline{v}_{2\mid k}^k(s,d)\leq{}&\mathbb{E}_{a\sim p_{\pi_1^k(s,d)}(\cdot|s),s'\sim p(\cdot|s,a)}\left[\overline{v}_3^k(s',\pi_2(s,d))-\overline{v}_{3\mid k}^k(s',\pi_2(s,d))\right]\\
&+L\mathbb{E}_{a\sim p_{\pi_1^k(s,d)}(\cdot|s)}\left[\min\{1,\beta^k(s,a,\delta)\}\right]\\
&+L\left[\min\{1,\beta_{\mathcal{D}}^k(s,\pi_2^k(s,d),\delta)\}\right]
\end{aligned} \tag{49}$$

Hence, by induction we have

$$\overline{v}_1^k(s,d)-\overline{v}_{1\mid k}^k(s,d)\leq L\mathbb{E}\left[\sum_{t=1}^L\min\{1,\beta_{\mathcal{D}}^k(s_t,d_t,\delta)\}+\sum_{t=1}^L\min\{1,\beta^k(s_t,a_t,\delta)\}|s_1=s,d_0=d\right] \tag{50}$$

where the expectation is taken over the MDP with policy $\pi^k$ under true transitions $P_{\mathcal{D}}$ and $P$. $\qquad\square$

**Lemma 5.** *Let $\mathcal{W}$ be a finite set and $c$ be a constant. For $k\in[K]$, suppose $\mathcal{T}_k=(w_{k,1},w_{k,2},\ldots,w_{k,H})$ is a random variable with distribution $P(.|w_{k,1})$, where $w_{k,i}\in\mathcal{W}$. Then,*

$$\sum_{k=1}^K\mathbb{E}_{\mathcal{T}_k\sim P(.|w_{k,1})}\left[\sum_{t=1}^H\min\{1,\frac{c}{\sqrt{\max\{1,N_k(w_{k,t})\}}}\}\right]\leq 2H|\mathcal{W}|+2\sqrt{2}c\sqrt{|\mathcal{W}|KH} \tag{51}$$

*with $N_k(w):=\sum_{j=1}^{k-1}\sum_{t=1}^H\mathbb{I}(w_{j,t}=w)$.*

*Proof.* The proof is adapted from Osband et al. (2013). We first note that

$$\begin{aligned}
\mathbb{E}\left[\sum_{k=1}^K\sum_{t=1}^H\min\{1,\frac{c}{\sqrt{\max\{1,N_k(w_{k,t})\}}}\}\right]={}&\mathbb{E}\left[\sum_{k=1}^K\sum_{t=1}^H\mathbb{I}(N_k(w_{k,t})\leq H)\min\{1,\frac{c}{\sqrt{\max\{1,N_k(w_{k,t})\}}}\}\right]\\
&+\mathbb{E}\left[\sum_{k=1}^K\sum_{t=1}^H\mathbb{I}(N_k(w_{k,t})>H)\min\{1,\frac{c}{\sqrt{\max\{1,N_k(w_{k,t})\}}}\}\right]\\
\leq{}&\mathbb{E}\left[\sum_{k=1}^K\sum_{t=1}^H\mathbb{I}(N_k(w_{k,t})\leq H)\cdot 1\right]\\
&+\mathbb{E}\left[\sum_{k=1}^K\sum_{t=1}^H\mathbb{I}(N_k(w_{k,t})>H)\cdot\frac{c}{\sqrt{N_k(w_{k,t})}}\right]
\end{aligned} \tag{52}$$

Then, we bound the first term of the above equation

$$\mathbb{E}\left[\sum_{k=1}^{K}\sum_{t=1}^{H}\mathbb{I}(N_k(w_{k,t})\le H)\right] = \mathbb{E}\left[\sum_{w\in\mathcal{W}}\{\#\text{ of times } w \text{ is observed and } N_k(w)\le H\}\right]$$

$$\le \mathbb{E}\left[|\mathcal{W}|\cdot 2H\right] = 2H|\mathcal{W}| \tag{53}$$

To bound the second term, we first define $n_\tau(w)$ as the number of times $w$ has been observed in the first $\tau$ steps, *i.e.*, if we are at the $t^{\text{th}}$ index of trajectory $\mathcal{T}_k$, then $\tau = t_k + t$, where $t_k = (k-1)H$, and note that

$$n_{t_k+t}(w) \le N_k(w) + t \tag{54}$$

because we will observe $w$ at most $t\in\{1,\dots,H\}$ times within trajectory $\mathcal{T}_k$. Now, if $N_k(w) > H$, we have that

$$n_{t_k+t}(w) + 1 \le N_k(w) + t + 1 \le N_k(w) + H + 1 \le 2N_k(w). \tag{55}$$

Hence we have,

$$\mathbb{I}(N_k(w_{k,t}) > H)(n_{t_k+t}(w_{k,t})+1) \le 2N_k(w_{k,t}) \implies \frac{\mathbb{I}(N_k(w_{k,t}) > H)}{N_k(w_{k,t})} \le \frac{2}{n_{t_k+t}(w_{k,t})+1} \tag{56}$$

Then, using the above equation, we can bound the second term in Eq. 52:

$$\mathbb{E}\left[\sum_{k=1}^{K}\sum_{t=1}^{H}\mathbb{I}(N_k(w_{k,t}) > H)\frac{c}{\sqrt{N_k(w_{k,t})}}\right] = \mathbb{E}\left[\sum_{k=1}^{K}\sum_{t=1}^{H}c\sqrt{\frac{\mathbb{I}(N_k(w_{k,t}) > H)}{N_k(w_{k,t})}}\right]$$

$$\overset{(i)}{\le} \sqrt{2}c\,\mathbb{E}\left[\sum_{k=1}^{K}\sum_{t=1}^{H}\sqrt{\frac{1}{n_{t_k+t}(w_{k,t})+1}}\right], \tag{57}$$

where (i) follows from Eq. 56.

Next, we can further bound $\mathbb{E}\left[\sum_{k=1}^{K}\sum_{t=1}^{H}\sqrt{\frac{1}{n_{t_k+t}(w_{k,t})+1}}\right]$ as follows:

$$\mathbb{E}\left[\sum_{k=1}^{K}\sum_{t=1}^{H}\sqrt{\frac{1}{n_{t_k+t}(w_{k,t})+1}}\right] = \mathbb{E}\left[\sum_{\tau=1}^{KH}\sqrt{\frac{1}{n_\tau(w_\tau)+1}}\right]$$

$$\overset{(i)}{=} \mathbb{E}\left[\sum_{w\in\mathcal{W}}\sum_{\nu=0}^{N_{K+1}(w)}\sqrt{\frac{1}{\nu+1}}\right]$$

$$= \sum_{w\in\mathcal{W}}\mathbb{E}\left[\sum_{\nu=0}^{N_{K+1}(w)}\sqrt{\frac{1}{\nu+1}}\right]$$

$$\le \sum_{w\in\mathcal{W}}\mathbb{E}\left[\int_{1}^{N_{K+1}(w)+1}\sqrt{\frac{1}{x}}dx\right]$$

$$\le \sum_{w\in\mathcal{W}}\mathbb{E}\left[2\sqrt{N_{K+1}(w)}\right]$$

$$\overset{(ii)}{\le} \mathbb{E}\left[2\sqrt{|\mathcal{W}|\sum_{w\in\mathcal{W}}N_{K+1}(w)}\right] \overset{(iii)}{=} \mathbb{E}\left[2\sqrt{|\mathcal{W}|KH}\right] = 2\sqrt{|\mathcal{W}|KH}, \tag{58}$$

where (i) follows from summing over different $w\in\mathcal{W}$ instead of time and from the fact that we observe each $w$ exactly $N_{K+1}(w)$ times after $K$ trajectories, (ii) follows from Jensen's inequality and (iii) follows from the fact that $\sum_{w\in\mathcal{W}}N_{K+1}(w) = KH$. Next, we combine Eqs 57 and 58 to obtain

$$\mathbb{E}\left[\sum_{k=1}^{K}\sum_{t=1}^{H}\mathbb{I}(N_k(w_{k,t}) > H)\frac{c}{\sqrt{N_k(w_{k,t})}}\right] \le \sqrt{2}c \times 2\sqrt{|\mathcal{W}|KH} = 2\sqrt{2}c\sqrt{|\mathcal{W}|KH} \tag{59}$$

Further, we plug in Eqs. 53 and 59 in Eq.52

$$\mathbb{E}\left[\sum_{k=1}^{K}\sum_{\tau=1}^{H}\min\{1, \frac{c}{\sqrt{\max\{1, N_k(w_{k,t})\}}}\}\}|\right] \leq 2H|\mathcal{W}|+2\sqrt{2}c\sqrt{|\mathcal{W}|KH} \tag{60}$$

This concludes the proof. $\qquad\square$

**Lemma 6.** *Let $\mathcal{W}$ be a finite set and $\mathcal{P}_t(\delta) := \{p : \forall w \in \mathcal{W}, ||p(.|w) - \hat{p}_t(.|w)||_1 \leq \beta_t(w, \delta)\}$ be a $|\mathcal{W}|$-rectangular confidence set over probability distributions $p^*(.|w)$ with $m$ outcomes, where $\hat{p}_t(.|w)$ is the empirical estimation of $p^*(.|w)$. Suppose at each time $\tau$, we observe an state $w_\tau = w$ and sample from $p^*(.|w)$. If $\beta_t(w, \delta) = \sqrt{\dfrac{2\log\left(\frac{t^7|\mathcal{W}|2^{m+1}}{\delta}\right)}{\max\{1, N_t(w)\}}}$ with $N_t(w) = \sum_{\tau=1}^{t}\mathbb{I}(w_\tau = w)$, then the true distributions $p^*$ lie in the confidence set $\mathcal{P}_t(\delta)$ with probability at least $1 - \frac{\delta}{2t^6}$.*

*Proof.* We adapt the proof from Lemma 17 in Jaksch et al. (2010). We note that,

$$\Pr(p^* \notin \mathcal{P}_t) \overset{(i)}{=} \Pr\left(\bigcup_{w \in \mathcal{W}} \|p^*(\cdot\,|\,w) - \hat{p}_t(\cdot\,|\,w)\|_1 \geq \beta_t(w, \delta)\right)$$

$$\overset{(ii)}{\leq} \sum_{w \in \mathcal{W}} \Pr\left(\|p^*(\cdot\,|\,w) - \hat{p}_t(\cdot\,|\,w)\|_1 \geq \sqrt{\frac{2\log\left(\frac{t^7|\mathcal{W}|2^{m+1}}{\delta}\right)}{\max\{1, N_t(w)\}}}\right)$$

$$\overset{(iii)}{\leq} \sum_{w \in \mathcal{W}}\sum_{n=0}^{t} \Pr\left(\|p^*(\cdot\,|\,w) - \hat{p}_t(\cdot\,|\,w)\|_1 \geq \sqrt{\frac{2\log\left(\frac{t^7|\mathcal{W}|2^{m+1}}{\delta}\right)}{\max\{1, n\}}}\right),$$

where (i) follows from the definition of the confidence set, *i.e.*, the probability distributions do not lie in the confidence set if there is at least one state $w$ in which $\|p^*(\cdot\,|\,w) - \hat{p}(\cdot\,|\,w)\|_1 \geq \beta_t(w, \delta)$, (ii) follows from the definition of $\beta_t(w, \delta)$ and a union bound over all $w \in \mathcal{W}$ and (iii) follows from a union bound over all possible values of $N_t(w)$. To continue, we split the sum into $n = 0$ and $n > 0$:

$$\sum_{w \in \mathcal{W}}\sum_{n=0}^{t} \Pr\left(\|p^*(\cdot\,|\,w) - \hat{p}_t(\cdot\,|\,w)\|_1 \geq \sqrt{\frac{2\log\left(\frac{t^7|\mathcal{W}|2^{m+1}}{\delta}\right)}{\max\{1, n\}}}\right)$$

$$= \sum_{w \in \mathcal{W}}\sum_{n=1}^{t} \Pr\left(\|p^*(\cdot\,|\,w) - \hat{p}_t(\cdot\,|\,w)\|_1 \geq \sqrt{\frac{2\log\left(\frac{t^7|\mathcal{W}|2^{m+1}}{\delta}\right)}{n}}\right)$$

$$+ \overbrace{\sum_{w \in \mathcal{W}} \Pr\left(\|p^*(\cdot\,|\,w) - \hat{p}_t(\cdot\,|\,w)\|_1 \geq \sqrt{2\log\left(\frac{t^7|\mathcal{W}|2^{m+1}}{\delta}\right)}\right)}^{\text{if } n=0}$$

$$\overset{(i)}{=} \sum_{w \in \mathcal{W}}\sum_{n=1}^{t} \Pr\left(\|p^*(\cdot\,|\,w) - \hat{p}_t(\cdot\,|\,w)\|_1 \geq \sqrt{\frac{2\log\left(\frac{t^7|\mathcal{W}|2^{m+1}}{\delta}\right)}{n}}\right) + 0$$

$$\overset{(ii)}{\leq} t|\mathcal{W}|2^m \exp\left(\log\left(-\frac{t^7|\mathcal{W}|2^{m+1}}{\delta}\right)\right) \leq \frac{\delta}{2t^6},$$

where (i) follows from the fact that $\|p^*(\cdot\,|\,w) - \hat{p}_t(\cdot\,|\,w)\|_1 < \sqrt{2\log\left(\frac{t^7|\mathcal{W}|2^{m+1}}{\delta}\right)}$ for non-trivial cases. More specifically,

$$\delta < 1,\, t \geq 2 \implies \sqrt{2log\left(\frac{t^7|\mathcal{W}|2^{m+1}}{\delta}\right)} > \sqrt{2\log(512)} > 2,$$

$$\|p^*(\cdot\,|\,w) - \hat{p}_t(\cdot\,|\,w)\|_1 \leq \sum_{i\in[m]} (p^*(i\,|\,s) + \hat{p}_t(i\,|\,w)) \leq 2, \tag{61}$$

and (ii) follows from the fact that, after observing $n$ samples, the $L^1$-deviation of the true distribution $p^*$ from the empirical one $\hat{p}$ over $m$ events is bounded by:

$$\Pr\left(\|p^*(\cdot) - \hat{p}(\cdot)\|_1 \geq \epsilon\right) \leq 2^m \exp\left(-n\frac{\epsilon^2}{2}\right) \tag{62}$$

**Lemma 7.** *Consider the following minimization problem:*

$$\begin{aligned} \underset{\boldsymbol{x}}{minimize} \quad & \sum_{i=1}^{m} x_i w_i \\ subject\ to \quad & \sum_{i=1}^{m} |x_i - b_i| \leq d,\ \sum_i x_i = 1, \\ & x_i \geq 0 \ \forall i \in \{1,\ldots,m\}, \end{aligned} \tag{63}$$

*where $d \geq 0$, $b_i \geq 0 \ \forall i \in \{1,\ldots,m\}$, $\sum_i b_i = 1$ and $0 \leq w_1 \leq w_2 \ldots \leq w_m$. Then, the solution to the above minimization problem is given by:*

$$x_i^* = \begin{cases} \min\{1, b_1 + \frac{d}{2}\} & \text{if } i = 1 \\ b_i & \text{if } i > 1 \text{ and } \sum_{l=1}^{i} x_l \leq 1 \\ 0 & \text{otherwise.} \end{cases} \tag{64}$$

*Proof.* Suppose there is $\{x_i';\ \sum_i x_i' = 1,\ x_i' \geq 0\}$ such that $\sum_i x_i' w_i < \sum_i x_i^* w_i$. Let $j \in \{1,\ldots,m\}$ be the first index where $x_j' \neq x_j^*$, then it's clear that $x_j' > x_j^*$.
If $j = 1$:

$$\sum_{i=1}^{m} |x_i' - b_i| = |x_1' - b_1| + \sum_{i=2}^{m} |x_i' - b_i| > \frac{d}{2} + \sum_{i=2}^{m} b_i - x_i' = \frac{d}{2} + x_1' - b_1 > d \tag{65}$$

If $j > 1$:

$$\sum_{i=1}^{m} |x_i' - b_i| = |x_1' - b_1| + \sum_{i=j}^{m} |x_i' - b_i| > \frac{d}{2} + \sum_{i=j+1}^{m} b_i - x_i' > \frac{d}{2} + x_1' - b_1 = d \tag{66}$$

Both cases contradict the condition $\sum_{i=1}^{m} |x_i' - b_i| \leq d$. $\qquad\square$

**Lemma 8.** *For the value function $V_{t|P_{\mathcal{D}},P}^{\pi}$ defined in Eq. 10, we have that:*

$$V_{t|P_{\mathcal{D}},P}^{\pi}(s,d) = c_{\pi_t(s,d)}(s,d) + \sum_{a\in\mathcal{A}} p_{\pi_t(s,d)}(a|s) \cdot \left(c_e(s,a) + \sum_{s'\in\mathcal{S}} p(s'\,|\,s,a) \cdot V_{t+1|P_{\mathcal{D}},P}^{\pi}(s',\pi_t(s,d))\right) \tag{67}$$

*Proof.*

$$V_{t|P_{\mathcal{D}},P}^{\pi}(s,d) \overset{(i)}{=} \bar{c}(s,d) + \sum_{s'\in\mathcal{S}} p(s',\pi_t(s,d)|(s,d)) V_{t+1|P_{\mathcal{D}},P}^{\pi}(s',\pi_t(s,d))$$

$$\overset{(ii)}{=} \sum_{a\in\mathcal{A}} p_{\pi_t(s,d)}(a\,|\,s)c_e(s,a) + c_c(\pi_t(s,d)) + c_x(\pi_t(s,d),d) + \sum_{s'\in\mathcal{S}}\sum_{a\in\mathcal{A}} p(s'\,|\,s,a)p_{\pi_t(s,d)}(a\,|\,s)V^\pi_{t+1|P_\mathcal{D},P}(s',\pi_t(s,d))$$

$$\overset{(iii)}{=} c_{\pi_t(s,d)}(s,d) + \sum_{a\in\mathcal{A}} p_{\pi_t(s,d)}(a|s)\cdot\left(c_e(s,a) + \sum_{s'\in\mathcal{S}} p(s'\,|\,s,a)\cdot V^\pi_{t+1|P_\mathcal{D},P}(s',\pi_t(s,d))\right), \tag{68}$$

where (i) is the standard Bellman equation in the standard MDP defined with dynamics 3 and costs 4, (ii) follows by replacing $\bar{c}$ and $p$ with equations 3 and 4, and (iii) follows by $c_{d'}(s,d) = c_c(d') + c_x(d',d)$. $\quad\square$

**Lemma 9.** $\min\{T, a+b\} \le \min\{T, a\} + \min\{T, b\}$ *for* $T, a, b \ge 0$.

*Proof.* Assume that $a \le b \le a + b$. Then,

$$\min\{T, a+b\} = \begin{cases} T \le a+b = \min\{T,a\} + \min\{T,b\} & \text{if } a \le b \le T \le a+b \\ T \le a+T = \min\{T,a\} + \min\{T,b\} & \text{if } a \le T \le b \le a+b \\ T \le 2T = \min\{T,a\} + \min\{T,b\} & \text{if } T \le a \le b \le a+b \\ a+b = \min\{T,a\} + \min\{T,b\} & \text{if } a \le b \le a+b \le T \end{cases} \tag{69}$$

$\square$

## C  Implementation of UCRL2 in finite horizon setting

---

**ALGORITHM 2:** Modified UCRL2 algorithm for a finite horizon MDP $\mathcal{M} = (\mathcal{S}, \mathcal{A}, P, C, L)$.

---

**Require:** Cost $C = [c(s,a)]$, confidence parameter $\delta \in (0,1)$.

1: $(\{N_k(s,a)\}, \{N_k(s,a,s')\}) \leftarrow \text{INITIALIZECOUNTS}()$

2: **for** $k = 1, \ldots, K$ **do**

3:    **for** $s, s' \in \mathcal{S}, a \in \mathcal{A}$ **do**

4:       **if** $N_k(s,a) \neq 0$ **then** $\hat{p}_k(s'|s,a) \leftarrow \dfrac{N_k(s,a,s')}{N_k(s,a)}$ **else** $\hat{p}_k(s'|s,a) \leftarrow \frac{1}{|\mathcal{S}|}$

5:       $\beta_k(s,a,\delta) \leftarrow \sqrt{\dfrac{14|\mathcal{S}|\log\left(\frac{2(k-1)L|\mathcal{A}||\mathcal{S}|}{\delta}\right)}{\max\{1, N_k(s,a)\}}}$

6:    **end for**

7:    $\pi^k \leftarrow \text{EXTENDEDVALUEITERATION}(\hat{p}_k, \beta_k, C)$

8:    $s_0 \leftarrow \text{INITIALCONDITIONS}()$

9:    **for** $t = 0, \ldots, L-1$ **do**

10:      Take action $a_t = \pi_t^k(s_t)$, and observe next state $s_{t+1}$.

11:      $N_k(s_t, a_t) \leftarrow N_k(s_t, a_t) + 1$

12:      $N_k(s_t, a_t, s_{t+1}) \leftarrow N_k(s_t, a_t, s_{t+1}) + 1$

13:    **end for**

14: **end for**

15: **Return** $\pi^K$

---

---

**ALGORITHM 3:** It implements EXTENDEDVALUEITERATION, which is used in Algorithm 2.

---

**Require:** Empirical transition distribution $\hat{p}(.|s,a)$, cost $c(s,a)$, and confidence interval $\beta(s,a,\delta)$.

1: $\pi \leftarrow \text{INITIALIZEPOLICY}(), \quad v \leftarrow \text{INITIALIZEVALUEFUNCTION}()$

2: $n \leftarrow |\mathcal{S}|$

3: **for** $t = T-1, \ldots, 0$ **do**

4:    **for** $s \in \mathcal{S}$ **do**

5:      **for** $a \in \mathcal{A}$ **do**

6:        $s_1', \ldots, s_n' \leftarrow \text{SORT}(v_{t+1})$                 # $v_{t+1}(s_1') \leq \ldots \leq v_{t+1}(s_n')$

7:        $p(s_1') \leftarrow \min\{1, \hat{p}(s_1'|s,a) + \frac{\beta(s,a,\delta)}{2}\}$

8:        $p(s_i') \leftarrow \hat{p}(s_i'|s,a) \,\forall\, 1 < i \leq n$

9:        $l \leftarrow n$

10:       **while** $\sum_{s_i' \in \mathcal{S}} p(s_i') > 1$ **do**

11:         $p(s_l') = \max\{0, 1 - \sum_{s_i' \neq s_l'} p(s_i')\}$

12:         $l \leftarrow l - 1$

13:       **end while**

14:        $q(s,a) = c(s,a) + \mathbb{E}_{s' \sim p}[v_{t+1}(s')]$

15:      **end for**

16:      $v_t(s) \leftarrow \min_{a \in \mathcal{A}}\{q(s,a)\}$

17:      $\pi_t(s) \leftarrow \arg\min_{a \in \mathcal{A}}\{q(s,a)\}$

18:    **end for**

19: **end for**

20: **Return** $\pi$

---

# D   Distribution of cell types and traffic levels in the lane driving environment

Table 1: Probability of cell types based on traffic level.

|        | road | grass | stone | car |
|--------|------|-------|-------|-----|
| no-car | 0.7  | 0.2   | 0.1   | 0   |
| light  | 0.6  | 0.2   | 0.1   | 0.1 |
| heavy  | 0.5  | 0.2   | 0.1   | 0.2 |

Table 2: Probability of traffic levels based on the previous row.

|        | no-car | light | heavy |
|--------|--------|-------|-------|
| no-car | 0.99   | 0.01  | 0     |
| light  | 0.01   | 0.98  | 0.01  |
| heavy  | 0      | 0.01  | 0.99  |

# E   Performance of the human and machine agents in obstacle avoidance task

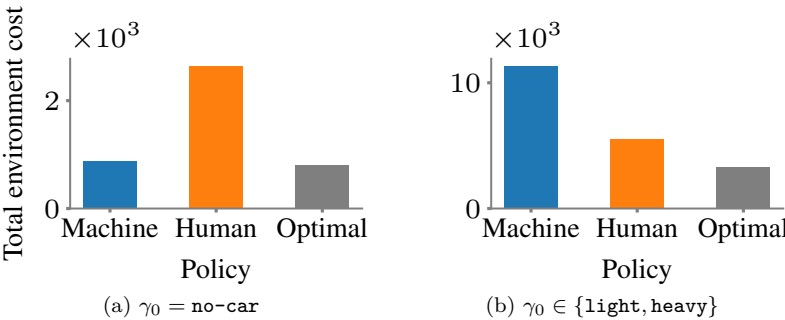

Figure 8: Performance of the machine policy, a human policy with $\sigma_{\mathbb{H}} = 2$, and the optimal policy in terms of total cost. In panel (a), the episodes start with an initial traffic level $\gamma_0 = \mathtt{no\text{-}car}$ and, in panel (b), the episodes start with an initial traffic level $\gamma_0 \in \{\mathtt{light}, \mathtt{heavy}\}$.

# F   The amount of human control for different initial traffic levels

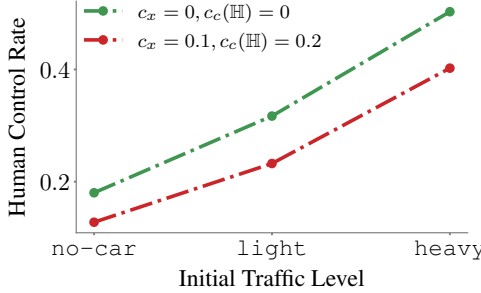

Figure 9: The amount of human control rate using UCRL2-MC switching algorithm for different initial traffic levels. For each traffic level, we sample 500 environment and average the human control rate over them. Higher traffic level results in more human control, as the human agent is more reliable in heavier traffic.

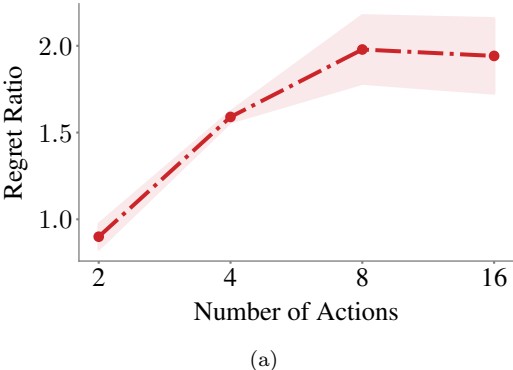 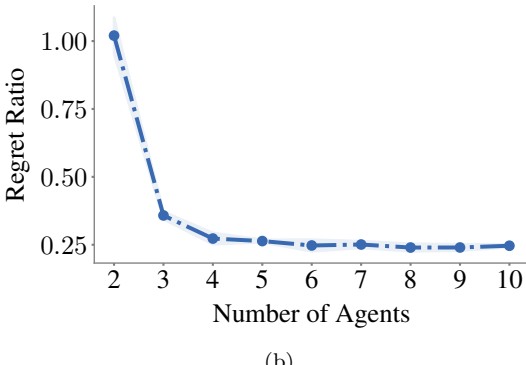

|       |       |
|-------|-------|
| (a)   | (b)   |

Figure 10: Ratio of UCRL2-MC regret to UCRL2 for (a) a set of action sizes and (b) different numbers of agents. By increasing the action space size, the performance of UCRL2-MC gets worse but remains within the same scale. In addition, UCRL2-MC outperforms UCRL2 in environments with a larger number of agents.

## G    Additional Experiments

In this section, we run additional experiments in the RiverSwim environment to investigate the effect of action space size and the number of agents in a team on the total regret.

### G.1    Action space size

To study the effect of action space size on the total regret, we artificially increase the number of actions by planning $m$ steps ahead. More concretely, we consider a new MDP, where each time step consists of $m$ steps of the original RiverSwim MDP, and the switching policy decides for all the $m$ steps at once. The number of actions in the new MDP increases to $2^m$, while the state space remains unchanged. We consider a setting with a single team of two agents with $p = 0$ and $p = 1$, i.e., one agent always takes action `right` and the other takes `left`. We run the simulations for 20,000 episodes with $m = \{1, 2, \cdots, 4\}$, i.e., with the action size of $2, 4, 8, 16$. Each experiment is repeated for 5 times. We compare the performance of our algorithm against UCRL2 in terms of total regret. Figure 10 (a) summarizes our results; The performance of UCRL2-MC gets worse by increasing the number of actions as the regret bound directly depends on the action size (Theorem 2). However, the regret ratio still remains within the same scale even after doubling the number of actions. One reason is that our algorithm only needs to learn *the actions taken by the agents* to find the optimal switching policy. If the agents' policies include a small subset of actions, our algorithms will maintain a small regret bound even in environments with huge action space. Therefore, we believe a more careful analysis can improve our regret bound by making it a function of agents' action space instead of the whole action size.

### G.2    Number of agents

Here, our goal is to examine the impact of the number of agents on the total regret achieved by our algorithm. To this end, we consider the original RiverSwim MDP (i.e., two actions) with a single team of $n$ agents, where we run our simulations for $n = \{3, 4, \cdots, 10\}$ and 20,000 episodes for each $n$. We choose $p$, i.e., the probability of taking action `right` for $n$ agents as $\{0, \frac{1}{n-1}, \cdots, \frac{n-2}{n-1}, 1\}$. As shown in Figure 10 (b), UCRL2-MC outperforms UCRL2 as the number of agents increases. This agrees with Theorem 2, as our derived regret bound mainly depends on the action space size $|\mathcal{A}|$, while the UCRL2 regret bound depends on the number of agents $|\mathcal{D}|$.

