# OpenReview forum: "Learning to Switch Among Agents in a Team via 2-Layer Markov Decision Processes"
_TMLR — Accepted by TMLR_

### Review · Reviewer_8f5p · 2022-05-12

**Summary Of Contributions:**

This paper addresses the problem of learning to switch between a set of fixed (but unknown) policies in a sequential setting. First, it formulates the problem as a 2-layer MDP, in which one layer is devoted to the policy selection, the other to the action selection. Then, it proposes a variant of the UCRL2 algorithm that deals with multiple confidence sets simultaneously, one for the unknown transition dynamics (as in standard UCRL2) and the others for the unknown policies. A regret analysis for this newly introduced algorithm is carried out both in a setting where a single team of agent (policies) interact with the environment, and in a setting where multiple teams of agents operate independently on the same environment. Finally, the paper provides an experimental validation in an obstacle-avoidance illustrative domain.

**Broader Impact Concerns:**

I think that the paper does enough to address the potential impact concerns. On the one hand, this paper is mainly theoretical. Moreover, the authors clearly report (at the end of the introduction) that adopting their approach for real world tasks, especially including human-machine collaborations, should require further analysis of the practical aspects.

**Requested Changes:**

- (Notation) The notation is not always easy to follow. I would suggest the authors to consider a more (visually) clear separation between the symbols that concern the switching-layer of the MDP, and the symbols that concern to the action-layer of the MDP. As a minor note, the paper is not always consistent in reporting the cardinality of the sets with the symbol $|\cdot|$.
- (Discussion of the setting) I would suggest the authors to include a deeper discussion on the relevance of the multiple-teams setting and its potential applications.
- (Experiments) The reported regret analysis suggests that UCRL-MC may struggle in domains with huge actions sets. Thus, I would consider an illustrative experiment to debunk this possibility. It could be also interesting to include experiments with larger teams of agents.

**Strengths And Weaknesses:**

*Strengths*
- (Relevance of the problem) The paper tackles the relevant problem of learning to switch between agents/policies online. This setting has important ramifications, especially the application in which human and artificial agents alternate control over a system;
- (Formulation) The proposed formulation of the switching problem into a 2-layer MDP is neat, and it allows for independent estimates of the agents' policies and the transition dynamics;
- (Clarity) The reported regret rates are easy to interpret, and they allow for a clear comparison with UCRL2 operating on a naive formulation of the switching problem into a 1-layer MDP;
- (Numerical validation) The numerical validation is compelling, and it supports the main theoretical claims;
- (Related Work) I am not particularly familiar with this area of research, but the related work section is well-written, and it seems to include the most relevant previous works (especially the lazy-MDP framework of Jacq et al., 2022).

*Weaknesses*
- (Regret rates) Unfortunately, handling the confidence sets independently for the agents' policies and the transition dynamics does not lead to a clear dominance in the regret rates over what one can achieve with standard UCRL.
	- In the single-team setting, the regret rate of UCRL-MC is at most on par with UCRL2, a conclusion that seems to be supported by the numerical experiments. However, the rate of UCRL-MC comes with its own drawbacks, such as a direct dependence with the cardinality of the action set A, which is instead avoided by UCRL2.
	- Instead, in the peculiar setting of multiple teams independently operating on similar environments, UCRL-MC seems to bring more clear benefits, especially if the number of such teams (N) is huge. While the reported experiment partially support this conclusion, the rates of UCRL2 and UCRL-MC still fall on a similar scale;
- (Techniques) The analysis seems to be mostly incremental w.r.t. standard UCRL2, which suggests a limited independent interest for the employed techniques;
- (Notation) The notation is a little convoluted, which makes it hard to follow some of the derivations.

*General comment*

This seems to be a valuable and interesting paper overall, even if the regret rate in the single-team setting is somewhat disappointing. I believe the paper could do more to support the relevance of the multiple-teams setting, as it is the one where their approach provides most of the benefit, but it also less clear what are the potential applications. Anyway, I think that the 2-layer formulation of the switching problem, along with the multiple-confidence algorithmic approach, could be valuable as an intermediate step towards an even more relevant setting, which is the switching problem over learning agents. Thus, I think this paper might be worthy of publication with minor changes.

---

> ### Author Response · Authors · 2022-06-21
> **Response to Reviewer 8f5p**
>
> Thank you for the detailed review and your time. Our response is as follows:
>
> **[Notation & cardinality of the sets]** We appreciate the reviewer’s concern about the difficulty of the Notation. However, since most symbols, including state $s$ and action $a$, are shared among switching and action layers, we decided not to change the Notation. Instead, to clarify the notion of 2-layer MDP, we added Figure 1 to illustrate the separation of the switching and action layers. Moreover, throughout the manuscript, we consistently reported the cardinality of the sets with the symbol $|⋅|$.
>
> **[Discussion of the setting]** We believe the multiple-teams setting may fit a variety of real applications; more prominently, think of a car manufacturer continuously collecting driving data from millions of human drivers wishing to learn different switching policies for each driver in order to implement a personalized semi-autonomous driving system. In the revised manuscript, we have included this real application to better motivate the multiple-teams setting at the beginning of Section 5 (Learning to Switch Across Multiple Teams of Agents). However, as explicitly noted at the end of Section 1 (Introduction), we would like to emphasize that a practical deployment of our methodology in a real application would require considering a wide range of additional practical aspects.
>
> **[Additional experiments]** To investigate the effect of the action space size on the regret bound, we included an additional experiment section in the appendix (section F). More specifically, we consider the RiverSwim environment (Strehl & Littman, 2008) with $m$-step ahead planning, which increases the action space size to $2^m$, and compared the results of our algorithm to UCRL2 for different values of $m$. The results show that, although the performance of UCRL2-MC decreases as we increase the number of actions, the regret remains within the same scale. We believe this is because our algorithm does not need to learn the dynamics of all actions but only the actions the agents would take. Therefore, a more careful analysis can result in a tighter regret bound, which will only depend on the agents’ action space instead of the full action size. We also examined the effect of increasing the number of agents and teams in the multiple teams of agents setting on the total regret. Our results show that UCRL2-MC significantly outperforms its problem agnostic version (UCRL2) when the number of teams is much larger (~100). Please see sections F.1, F.2, and F.3 for more details. We hope these additional experiments will address your concern about the regret rates of UCRL2-MC in comparison to the standard UCRL2 algorithm.

---

### Review · Reviewer_CWXS · 2022-05-16

**Summary Of Contributions:**

The paper focuses on formalizing the problem of switching control policies between multiple agents in a team. A 2-layer MDP is used to frame this problem and an on-line learning algorithm with provable guarantees (UCRL2-MC) is proposed. The proposed algorithm uses upper confidence bounds on agent policies as well as environment transition probabilities. Shared confidence bounds allow better regret bound than problem agnostic algorithms. The improvements are demonstrated on a pair of toy problems.

**Broader Impact Concerns:**

Not required.

**Requested Changes:**

The title is too broad and does not correctly characterize the contributions of the paper. A single paper obviously doesn't solve all the problems and baby steps are fine, but there are many ways to frame the problem of switching agent policies and 2-layer MDP choice in the paper is just one among many.
The current limitations of the work could be much better stated as noted above.

Given the interests laid out in future work, while the means are quite different, [1] might be of interest given the interaction of switching between policies specializing in different transition dynamics settings using ane estimate of transition dynamics information.

[1] https://arxiv.org/abs/1903.01567

**Strengths And Weaknesses:**

## Strengths

Majority of work in the community is focused on proposing alogrithms or systems aiming for full autonomy in some way. However, for real world applications often human are required in the loop and one needs an ensemble of policies. Relatively less work has focused on this problem, so it's good to have this paper trying to formalize some aspects of such settings.
The paper is clearly laid out and well written. The choice of 2-layer MDP is well defended and is quite insightful in understanding the problem decomposition. While I am not the best judge for confirming whether the proof is correct, it does seem alright to me and given the framing choices, the improvements in regret bounds make sense. The simple experiments also have their setup quite clearly laid out with clear descriptions and help support the case for their effectiveness at least theoretically.

## Weaknesses

The paper doesn't actually solve many important challenges for their solution to be of actual practical interest (especially for human policies mixed in). Concept of time, especially human reaction time is completely missing from the discussion, with Markovian policies assumed to be a decent abstraction.
Similarly while given the theoretical nature of majority of the work, some discussion of the general reward design principles in this setting is warranted.

---

> ### Author Response · Authors · 2022-06-21
> **Response to Reviewer CWXS**
>
> We thank the reviewer for their time and comments. Here is our response:
>
> **[Human reaction time]** We acknowledge that, in a real application with human and machine agents, there may be scenarios in which one needs to explicitly model (human) agents' reaction times. We have explicitly discussed this caveat in the revised manuscript at the end of Section 1 (Introduction).
>
> **[General reward design principles]** The specific choice of environment, control, and switching costs are application dependent. We have clarified this in Footnote 4 in Section 3 (Switching Control Among Agents as a 2-Layer MDP).
>
> **[Too broad title]** We have changed the paper's title to "Learning to Switch Among Agents in a Team via a 2-Layer Markov Decision Process".
>
> **[Citation to Wu et al.]** We have added a citation to [1] https://arxiv.org/abs/1903.01567 in Section 7 (Conclusions and Future Work).

---

### Review · Reviewer_EUMX · 2022-06-06

**Summary Of Contributions:**

In this work, the authors proposed a MDP framework to study an RL problem that is less autonomous. In particular, they restrcit the problem formulation into one that allow allows the RL agent to take some of the actions and
leave the remaining ones to human agents, and argue that the resulting performance may be better than the performance
of the fully autonomous counterpart. To solve this problem defined by the authors, they extended the UCRL-based algorithm that uses upper confidence bounds on the agents’ policies and the environment’s transition probabilities to find a sequence of switching policies in a final horizon two-layer MDP. Same as standard UCRL-style analysis, the total regret of the derived algorithm  algorithm with respect to the optimal switching policy is sub-linear in the number of learning steps, and with multiple teams of agents operating in a similar environment, the proposed algorithm can also benefit from maintaining shared confidence bounds for the environments’
transition probabilities to have a better regret bound than traditional problem-agnostic MDP algorithms.

**Broader Impact Concerns:**

It's a paper focused on deriving new RL theories and algorithms on a specific two-layer MDP, in my opinions there are no particular ethical concerns.

**Requested Changes:**

Main changes to improve the paper include addressing the aforementioned questions and making the motivation of this paper more compelling, in terms of i) comparing this approach with the option or option critic framework, ii) the rationale behind having P_D and P decomposed this way, especially on the reasons on instead of just using the realized actions for planning, what is the additional value to keep track of how the actual actions are generated by the particular controllers, iii) similar to the criticisms of options, when executing the proposed UCRL2-like algorithm, where both the switching policy pi and "low-level policy"  p_d are minimized (in say (7) or (11)), how can one avoid having all the policies degenerate into a single flat policy that is the same as solving the original MDP problem, iv) what is the main motivation of extending the original algorithm to switch across multiple teams of agents as in Sec 5? , v) more involved experiments besides a simple grid-based domain as a  proof-of-concept.

Therefore, at this point I'd not vote for acceptance of this paper until the above comments are addressed.

**Strengths And Weaknesses:**

Strengths:
In this work the authors studied an alternative form of MDP in which the RL agent only controls the switches between particular human/non-autonomous agents that controls the generation of "low-level actions", for which the RL agent cannot be modified by the RL agent. This may relate to a specific instance of hierarchical RL in which the switching policy controls the high-level, marco action where the low-level actions are generated by policies that in this work needs to be fixed and cannot be fine-tuned. In general making the action space of RL more structural and introducing more priors into simplifying the original problem-agnostic RL framework is an important direction to make RL more efficient.

Weaknesses:
There are four questions that I hope the paper can address:
i) I understand that in this work the authors specifically fixed the "low-level" policy and formulate the problem of RL as designing the high-level switching policy. In term of novelty, how does that compare with the option-RL framework which has been around in the community for long? Can any of the option terminology be used here?
ii) One difference (that I appreciate the authors' pointed out) between existing "policy mixing" approach with the proposed approach is planning upon realized actions generated by low-level agents, whereas in many existing approaches the high-level agent only plans to choose which low-level agent, but not on choosing the realized actions. This formulation is similar to the "mixture-of-expert" approach, for example mentioned in Section 6 of this paper: https://arxiv.org/abs/2206.00059. I agree that we should augment the state space with the realized actions (which corresponds to the P_D environment introduced in this work). However, one question I have is why should we keep track of the evolution of the policy indexes in the state and state transition of the MDP, on top of just keep track of the transition of realized actions via P_D? At the end different agents may generate identical actions but our system may not necessarily care which agents generate the these actions.  Is there a Bellman optimality condition to show that the augmented state space chosen by the authors is "sufficient" and "necessary"?
iii) In the UCRL algorithm (in say (7) or (11)), it seems the environment is also minimized (due to the optimistic MDP formulation of UCRL). But in this case, this is exactly like optimizing both high-level policy \pi that generates macro actions and low-level policies that generate micro actions in hierarchical MDPs. Similar to most criticisms of these approaches in the literature (of options or hierarchical RL), how does the UCRL approach ensure that the training procedure does not degenerate into picking a flat, non-hierarchical policy?
iv) The motivation of extending the algorithm to learning a policy to switch multiple teams of agents is unclear. Are the authors claiming this extension to be more data-efficient for the multi-task RL setting? If so, besides extending the UCRL bound, more analysis and studies in the multi-task problem formulation are needed. Otherwise, this section 5 seems like an incremental addition and may be a distraction of the main message.
v) To better illustrate the effectiveness of the algorithms besides a simple grid-based driving experiment, it'd be good to see how this proposed algorithm performs in more involved numerical experiments. Also, even in the simple domain where the assumptions of the theorem can be verified, it'd be good to see if the numerical regret bound matches with the ones proven in the theoretical results, to understand the tightness of the theory bounds.

---

> ### Author Response · Authors · 2022-06-21
> **Response to Reviewer EUMX**
>
> We appreciate the reviewer's time and detailed comments. Here, we address questions (i) to (iii). We respond to questions (iv) and (v) in another comment.
>
> **[Comparing this approach with the option or option critic framework]**
>
> The notion of macro-actions and micro-actions in the options framework is indeed similar to our framework's switching policy and agent policy. Thus, one could think of using option terminology to formulate our problem.
> However, the option framework is designed to address different levels of temporal abstraction in RL by defining macro-actions that correspond to sub-tasks (skills) [Sutton et al. 1999]. In our problem, each agent is not necessarily optimized to act for a specific task or sub-goal but the whole environment/goal. Also, in our problem, we do not necessarily have control over all agents to learn the optimal policy for each of them, while in the option framework, a primary direction is to learn optimal options for each sub-task. In other words, even though we can mathematically refer to each agent policy as an option, they are not conceptually the same.
>
> The main novelty of our approach is to learn agent policies and environment transition probabilities separately and reuse the knowledge of the environment to get a more sample-efficient algorithm. A naive algorithm to learn optimal switching policy based on the option framework, which is equivalent to finding the optimal policy over a set of fixed options, does not reuse the environment transition probabilities and results in worse regret bounds. We explicitly show this by comparing the regret of our algorithm with the naive UCRL2 algorithm.
>
> We have added a comparison of our framework to the options framework in the related work section.
>
> **[The rationale behind having $P_D$ and $P$ decomposed]**
>
> First, note that our method has a key difference from the "mixture-of-expert" approach in https://arxiv.org/abs/2206.00059 [Chow et al. 2022]. In their method, they augment their state space with the realized actions and decide based on both the current state and the actions suggested by the experts. However, in our method, we cannot observe an agent's action before giving control to it. Therefore, we must learn their action policy while optimizing the switching policy. To this end, we decouple $P_D$, which corresponds to the action policy of unknown agents, and $P$ (transition probabilities of the environment) and learn them separately. Second, the advantage of keeping track of $P$ is that we can reuse the knowledge of environment transitions in the case of multiple teams of agents operating in the same environment. As discussed in Section 5, learning $P$ separately results in better regret bounds than naive approaches that only plan to choose low-level agents without decoupling $P_D$ and $P$ (See Corollary 3 and the discussion below). Finally, augmenting the state space with $\mathcal{D}$, which corresponds to the set of agents and not the realized actions, addresses the cost of switching $c_x$ (see eq. 1). Since the switching cost depends on both the current and previous agents, we need to keep track of past agents to minimize $c_x$. Also, please see Theorem 1, eq. (12) for the Bellman optimality formula of the augmented state space.
>
> **[How can one avoid having all the policies degenerate into a single flat policy that is the same as solving the original MDP problem]**
>
> Our approach uses the _optimism in the face of uncertainty_ principle to address the exploration-exploitation trade-off and learn $P$ and $P_D$. It is true that we optimize for both the environment and the agent policies. However, as we observe more samples from $P$ and $P_D$, the $L^1$ confidence intervals defined in Section 4 shrink and converge to the true $P^*$ and $P^*_{D}$ values. Therefore, in the limit, $\mathcal{P}^k$ and $\mathcal{P}^k_{\mathcal{D}}$ will only contain the true $P^*$ and $P^*_{D}$, and the minimization in (7) or (11) will be only over switching policies. Note that our algorithm does not choose flat, non-hierarchical policies by design since we only optimize the switching policy and have no control over agent policies. In other words, as long as the agents do not have the same action policy, our solution is inherently a hierarchical policy.

---

> > ### Author Response · Authors · 2022-06-21
> > **Response to Reviewer EUMX-Questions (iv) and (v)**
> >
> > **[What is the main motivation for extending the original algorithm to switch across multiple teams of agents as in Sec 5?]**
> >
> > In the real world, RL environments are not typically designed solely for a single run of a task. For example, in a car driving scenario, we observe multiple vehicles acting in the same environment. A key advantage of our algorithm over naive approaches such as UCRL2 is separating agents' action policies from environment transition probabilities, where the latter can be reused in similar tasks. In section 5, we consider one such possible scenario, where we have multiple teams of agents (e.g., cars) acting in the same environment (e.g., the same city). We avoid the term "multi-task RL" since the task, i.e., the environment transition probabilities and the reward function remains unchanged, but each team's set of agents is different. We claim that using our approach will result in a lower regret bound and thus a more data-efficient algorithm than naive approaches that only plan to choose low-level agents without separating the environment and agents. We support our claim both theoretically (Corollary 3 and the discussion below it) and empirically (Section 6.5). Note that our formulation of multiple teams of agents in Section 5 is one possible scenario in which the knowledge of environment transitions can be useful. Another potential scenario is multi-task RL with predictable changes to environment transitions. We leave such cases for future work.
> >
> > **[More involved experiments besides a simple grid-based domain as a proof-of-concept.]**
> >
> > We added a section on additional experiments to examine the impact of action size, the number of agents, and the number of teams on the total regret for a different RL task. Please see Section F in the appendix for details. We appreciate the reviewer's opinion/suggestions on other possible scenarios or any particular experimental setup.

---

### Comment · Action_Editors · 2022-06-07
**Reviews are out. Authors will have 2 weeks to discuss with reviewers.**

Thanks for the 3 reviewers for putting in their time to write a detailed review of this work.

Dear Authors,

Now that the reviews are out, please review the comments from all three reviewers, in particular, the points deemed to be weakness of the papers, and also requests for changes.

The goal of this discussion period is for the reviewers to gather all the information they need to be comfortable submitting a decision recommendation for this submission, in two weeks time from today.

**[TO DO] Respond to each review, write overall rebuttal, as soon as you can.**

You can carry out the discussions however you like, but the usual format that people have been doing this is for the authors to individually respond to each review, and also write a general "overall" rebuttal summarizing the 3 reviews, and what they have done to address the concerns (which coincides with an update of the pdf to address any concerns). During the rebuttal, you can refer to the original pdf, or the updated pdf, but to avoid miscommunication, please carefully refer to exactly which version of the pdf you are referring to during the discussions.

Good luck!

---

### Decision · Action_Editors · 2022-07-13

**Recommendation:** Accept as is

**Comment:**

The work formalizes the problem of learning to switch between a set of fixed control policies (agents) in a team, rather than going about solving a large policy space. A two-layer MDP is used to frame this problem, where one layer is for policy selection, and the other is to select actions. A learning algorithm with provable guarantees (UCRL2-MC) is proposed (a variant of UCRL2 algorithm), which uses upper confidence bounds on agent policies as well as environment transition probabilities. A regret analysis is carried out both in a setting where a single team of agents interact with the environment, and in a setting where multiple teams of agents operate independently in the same environment. They conduct a pair of experiments to demonstrate their approach.

Many issues were raised in the review period, during which the authors clarified the relationship of this work with existing frameworks (e.g. options) and methodologies (mixture-of-experts), which was raised by reviewer EUMX. The authors also better scoped claims and contributions (which helped address the issues raised by reviewer CWXS). Additional experiments (suggested by 8f5p) were conducted to investigate the effect of the action space size on the regret bound, and also to show benefits over existing approaches that they build on (e.g. UCRL2). Some were then noted in the main article (from the Appendix section) for clarity.

Reviewer EUMX mentions in the rebuttal discussion that several questions such as degeneration and distinction from existing work have been clarified, but the main issue they have with the work is that the experimental justification of this work is still on the weaker side. EUMX is not proposing a complex task, but one that is more related to the theory developed in the work to demonstrate clearly how the approach affects the performance in practice, resulting in a leaning reject recommendation from the

Other reviewers, such as CWXS, didn't mind the use of the 'toy problems' (quoted in their review) to demonstrate the improvements offered by (UCRL2-MC), since a main contribution is the theoretical guarantees that had been provided (which to some extent, offset the weaker set of experiments). Reviewer 8f5p initially had issues and concerns with the experiments too, but these were addressed during the rebuttal period, and reviewer CWXS was left satisfied with all of their concerns addressed.

While I agree with EUMX that the experiments are lacking when compared to many experimental papers (which are the majority of RL literature these days), and that the approach can be seen as incremental with modest novelty (which will likely lead to rejection at typical so called 'top' ML conferences), I also acknowledge that the contribution of this work also comes from the theoretical analysis, and that the other 2 reviewers were satisfied that the experiments (simple as they may be), were convincing enough to support the claims made (however modest they may be), and that they also complement the theoretical contributions. From the reviewers' comments, I also believe that there would at least be *some* individuals in TMLR's audience that would like to know the findings of this paper, so based on these points, my decision is to recommend **acceptance** of the work, as they satisfy TMLR's acceptance criteria:

---
The acceptance decision for a submission is based on the answers to the following questions:

1. Are the claims made in the submission supported by accurate, convincing and clear evidence?
2. Would at least some individuals in TMLR's audience be interested in knowing the findings of this paper?

Papers should be accepted if they meet the criteria, even if the contribution or significance of the work is modest.

---

That being said, I believe EUMX’s comments are still helpful, and their valuable feedback has helped shape and improve the work to its current acceptable level. I encourage the authors to extend their effort to include more practical examples as suggested, as that is still required to make this work broadly applicable beyond a niche research community.